# The genetic architecture of membranous nephropathy and its potential to improve non-invasive diagnosis

Jingyuan Xie et al.[#]

Membranous Nephropathy (MN) is a rare autoimmune cause of kidney failure. Here we report a genome-wide association study (GWAS) for primary MN in 3,782 cases and 9,038 controls of East Asian and European ancestries. We discover two previously unreported loci, *NFKB1* (rs230540, OR = 1.25, $P = 3.4 \times 10^{-12}$) and *IRF4* (rs9405192, OR = 1.29, P = $1.4 \times 10^{-14}$), fine-map the *PLA2R1* locus (rs17831251, OR = 2.25, $P = 4.7 \times 10^{-103}$) and report ancestry-specific effects of three classical HLA alleles: *DRB1\*1501* in East Asians (OR = 3.81, $P = 2.0 \times 10^{-49}$), *DQA1\*0501* in Europeans (OR = 2.88, $P = 5.7 \times 10^{-93}$), and *DRB1\*0301* in both ethnicities (OR = 3.50, $P = 9.2 \times 10^{-23}$ and OR = 3.39, $P = 5.2 \times 10^{-82}$, respectively). GWAS loci explain 32% of disease risk in East Asians and 25% in Europeans, and correctly re-classify 20–37% of the cases in validation cohorts that are antibody-negative by the serum anti-PLA2R ELISA diagnostic test. Our findings highlight an unusual genetic architecture of MN, with four loci and their interactions accounting for nearly one-third of the disease risk.

[#]A full list of authors and their affiliations appears at the end of the paper.

Membranous Nephropathy (MN) is a rare cause of kidney failure, manifesting as nephrotic syndrome with a peak incidence between 30 and 50 years of age[1]. The landmark discoveries of pathogenic antibodies against neutral endopeptidase in antenatal MN[2], and anti-phospholipase A2 receptor (PLA2R) antibodies in adult MN[3] have established MN as the disease of autoantibodies directed against podocyte antigens. Several studies have confirmed the presence of autoantibodies against PLA2R in ~60–70% of cases of primary MN[4], with another 3–5% potentially explained by antibodies against thrombospondin type 1 domain-containing 7A[5].

Previous genome-wide association study (GWAS) for MN conducted in 75 French, 146 Dutch and 335 British cases genotyped with low resolution arrays identified impressively strong associations of the HLA region and the PLA2R1 locus encoding the dominant antigen in MN[6]. These findings suggest that genetic variation controls the immunogenicity and/or expression level of the PLA2R auto-antigen, as well as the production of anti-PLA2R autoantibodies in individuals with a permissive HLA haplotype. However, specific causal alleles underlying GWAS associations have not yet been mapped at high resolution. Moreover, prior GWAS was limited to Europeans, and the reported associations have not been examined comprehensively across different ethnicities. Lastly, because of small sample size, the prior study might have missed additional disease relevant loci.

Herein, we report a genetic study of primary MN involving 12,820 individuals (3782 biopsy-documented cases and 9038 ancestry-matched controls), across nine cohorts of East Asian and European ancestries. The composition of our cohorts reflects the demographics of the centres that have collected DNA samples for genetic studies of this rare disease over the past 15 years. By using high resolution arrays with genome-wide imputation and over 7-fold increase in sample size compared to the prior GWAS, we discover two previously unreported genome-wide significant risk loci for MN and perform high resolution mapping and ethnicity-specific analyses of the known loci.

We describe an unusual genetic architecture of MN, with four loci and their genetic interactions accounting for nearly one-third of the disease risk. Our study implicates dysregulation of NFKB1 and IRF4 genes in the disease pathogenesis, providing genetic support for potential targeting of the NF-κB and interferon signalling pathways in primary MN. We also refine ethnicity-specific effects at the HLA locus, defining DRB1*1501 as a major risk allele in East Asians, DQA1*0501 in Europeans, and DRB1*0301

in both ethnicities. We describe a risk haplotype at the PLA2R1 locus that has a regulatory function and exhibits strong genetic interactions with the HLA-DRB1 risk alleles. Lastly, we calculate a genetic risk score (GRS) based on these findings which, when used in combination with a serum anti-PLA2R ELISA (a serologic test for MN currently in clinical use), shows superior performance in discriminating cases and controls than the ELISA or GRS alone. We validate the performance of this combined risk score (CRS) in external validation cohorts. Our results demonstrate that a combined serum-genetic test can potentially be used to establish a new diagnosis of primary MN, obviating the need for a high risk kidney biopsy procedure in the majority of cases.

## Results

**Study design.** Our study involved nine case-control cohorts, including four East Asian cohorts of 4841 individuals (1632 primary MN cases and 3209 controls) and five European cohorts of 7979 individuals (2150 primary MN cases and 5829 controls). Eight cohorts were genotyped with high density SNP arrays, imputed using the latest whole genome sequence reference panels, and meta-analyzed genome-wide, and the top 46 loci selected based on $P < 5 \times 10^{-5}$ were tested by targeted genotyping in the ninth cohort (Supplementary Table 1). The summary of study cohorts, genotyping methods, and ancestry-specific imputation panels is provided in Table 1.

All cases used in this study were defined by a kidney biopsy diagnosis of idiopathic MN and any suspected secondary cases due to drugs, malignancy, infection, or autoimmune disease were excluded. With the exception of the German Chronic Kidney Disease (GCKD) cohort, all controls used for discovery involved healthy population controls and any individuals with a known diagnosis of kidney disease were excluded. The GCKD cohort was drawn entirely from a prospective observational study of patients with CKD and consisted of biopsy-defined cases and controls for whom CKD etiology was clearly assigned to a non-MN cause, as previously described[7].

All genome-wide significant loci ($P < 5 \times 10^{-8}$) were refined by cohort-stratified stepwise conditional analyses to define independently associated haplotypes. We also analyzed classical HLA alleles and all common amino acid polymorphisms at class I and class II genes imputed at high resolution. We performed detailed genomic annotations and explored epistatic effects for significant loci. Based on significant GWAS loci, we designed a GRS for MN

**Table 1 Baseline characteristics of participants in the discovery and replication cohorts.**

| Cohort | Ancestry | No. of cases | No. of controls | Total | Genotyping platform | Imputation reference population panel |
|---|---|---|---|---|---|---|
| Asian Cohorts | | | | | | |
| Chinese Discovery | East Asian | 561 | 904 | 1465 | Zhonghua-8 chip (Illumina) | 1000G Phase 3 East Asians |
| Korean Discovery | East Asian | 164 | 708 | 872 | MEGA chip (Illumina) | 1000G Phase 3 East Asians |
| Japanese Discovery | East Asian | 81 | 358 | 439 | MEGA chip (Illumina) | 1000G Phase 3 East Asians |
| Chinese Replication | East Asian | 826 | 1239 | 2065 | KASP (targeted) | – |
| All Asian | | 1632 | 3209 | 4841 | | |
| European Cohorts | | | | | | |
| European Discovery 1 | European | 611 | 1246 | 1857 | MEGA chip (Illumina) | 1000G Phase 3 Europeans |
| European Discovery 2 | European | 1045 | 1094 | 2139 | MEGA chip (Illumina) | 1000G Phase 3 Europeans |
| Turkish Discovery | European | 254 | 336 | 590 | MEGA chip (Illumina) | 1000G Phase 3 Europeans |
| Sardinian Discovery | European | 93 | 1498 | 1591 | OmniExpress (Illumina) | 1000G Phase 3 Europeans |
| GCKD Discovery | European | 147 | 1655[a] | 1802 | Omni2.5Exome (Illumina) | HRC 1.1 |
| All European | | 2150 | 5829 | 7979 | | |
| All Participants | | 3782 | 9038 | 12,820 | | |

[a]GCKD participants with chronic kidney disease etiology assigned to a non-MN cause (see Methods).

and performed its validation in external cohorts, including the three previously published European GWAS cohorts[6] and the Nephrotic Syndrome Study Network (NEPTUNE) study[8].

The descriptions of all cohorts including ancestry analyses and details of statistical approaches are provided in Methods, Supplementary Methods, and Supplementary Figs. 1 and 2.

**Genome-wide association**. The results of combined genome-wide meta-analyses are summarized in Fig. 1 and Table 2, with more information provided in Supplementary Table 1 and Supplementary Figs. 3 and 4. We discovered two novel genome-wide significant loci: a locus on chromosome 4q24 encoding *NFKB1* (rs230540, OR = 1.25, Meta-analysis $P = 3.4 \times 10^{-12}$) and a locus

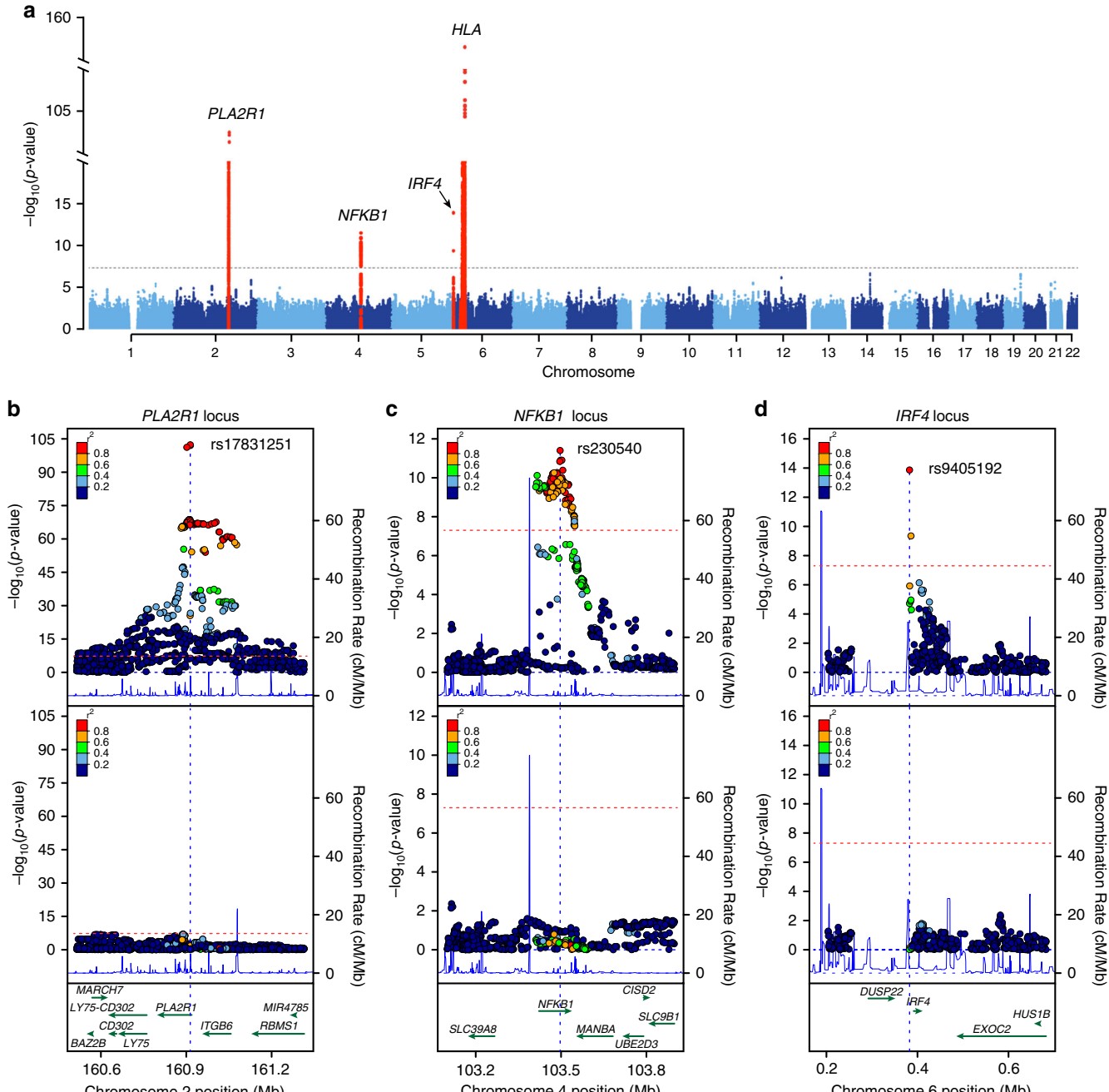

**Fig. 1 Manhattan and regional plots for non-HLA loci for the combined meta-analysis of all MN cohorts. a** The results of the combined meta-analysis across all cohorts; the dotted horizontal line indicates a genome-wide significance threshold ($\alpha = 5 \times 10^{-8}$); the *y*-axis is truncated twice to accommodate large peaks over *PLA2R1* and *HLA* loci; genome-wide-significant loci highlighted in red; **b** Regional plot for the *PLA2R1* locus; the upper panel shows unconditioned meta-results, the lower panel depicts meta-results after conditioning for the top SNP (rs17831251). **c** Regional plot for the *NFKB1* locus; the upper panel corresponds to unconditioned results; the lower panel shows meta-results after controlling for rs230540. **d** Regional plot for the *IRF4* locus; the upper panel corresponds to unconditioned results; the lower panel shows meta-results after controlling for rs9405192. The *x*-axis denotes genomic location (hg19 coordinates), left *y*-axis represents −log *P* values for association statistics, right *y*-axis represents average recombination rates based on HapMap-III reference populations combined (blue line). In the conditional analyses, we conditioned on the top SNP in each individual cohort, then meta-analyzed conditioned summary statistics as described in the Methods.

**Table 2 Effect estimates for top GWAS SNPs by ethnicity and combined across all cohorts.**

| Locus | SNP | Risk allele | E. Asian case freq. | E. Asian control freq. | E. Asian OR (95% CI) | E. Asian $P$-value | European case freq. | European control freq. | European OR (95% CI) | European $P$-value | Combined OR (95% CI) | Combined $P$-value |
|---|---|---|---|---|---|---|---|---|---|---|---|---|
| PLA2R1 | rs17831251 | C | 0.85 | 0.70 | 2.81 (2.48–3.17) | $3.5 \times 10^{-61}$ | 0.76 | 0.61 | 1.98 (1.81–2.17) | $4.7 \times 10^{-48}$ | 2.25 (2.09–2.42) | $4.7 \times 10^{-103}$ |
| NFKB1 | rs230540 | C | 0.43 | 0.35 | 1.24 (1.14–1.36) | $1.8 \times 10^{-6}$ | 0.35 | 0.32 | 1.25 (1.14–1.36) | $7.8 \times 10^{-7}$ | 1.25 (1.17–1.33) | $3.4 \times 10^{-12}$ |
| IRF4 | rs9405192 | G | 0.51 | 0.42 | 1.40 (1.28–1.53) | $8.8 \times 10^{-14}$ | 0.73 | 0.69 | 1.18 (1.07–1.29) | $6.6 \times 10^{-4}$ | 1.29 (1.21–1.37) | $1.4 \times 10^{-14}$ |
| HLA | rs9271573 | A | 0.60 | 0.35 | 2.97 (2.69–3.28) | $3.7 \times 10^{-102}$ | 0.62 | 0.44 | 2.06 (1.89–2.25) | $1.8 \times 10^{-60}$ | 2.41 (2.26–2.57) | $2.7 \times 10^{-154}$ |

on chromosome 6p25.3 encoding *IRF4* (rs9405192, OR = 1.29, Meta-analysis $P = 1.4 \times 10^{-14}$). We also confirmed strong and highly significant associations at the previously described loci, including chromosome 2q24.2 encoding *PLA2R1* (rs17831251, OR = 2.25, Meta-analysis $P = 4.7 \times 10^{-103}$) and 6p21.32 encoding *HLA-DQA1/DRB1* genes (rs9271573, OR = 2.41, Meta-analysis $P = 2.7 \times 10^{-154}$).

Conditional analyses of the three non-HLA loci revealed that each signal is explained by a single SNP in each cohort, suggesting a single shared risk haplotype per locus in East Asian and European populations (Fig. 1b–d). To further test if the causal variants at these loci are likely shared between Europeans and East Asians, we performed 99% credible set analyses using summary statistics for each ancestry-defined subgroup and compared them with credible sets derived from the trans-ethnic meta-analysis. We confirmed that the predicted causal variants derived from the trans-ethnic analysis were largely overlapping with ancestry-specific results (Supplementary Fig. 5). In contrast, stepwise conditional analyses of SNPs at the HLA region revealed a complex pattern of association, with at least three independently genome-wide significant SNPs explaining the signal across all cohorts (Supplementary Table 2).

Given the complexity of the association signal at the HLA locus and known differences in linkage disequilibrium (LD) patterns by ancestry, we performed additional analyses of this region separately in East Asians and Europeans. In the conditional analyses of the East Asian cohorts, only two independently associated SNPs explained the entire signal at this locus (rs9269027 and rs1974461). In Europeans, stepwise conditional analyses revealed three independently associated genome-wide significant SNPs (rs9271541, rs9265949, and rs2858309), suggesting a more complex pattern of association (Supplementary Table 2). In both ethnicities, the top signal centred on *HLA-DRB1* and *DQA1* genes (Fig. 2a, b).

**Classical HLA alleles and amino acid polymorphisms.** We next imputed classical HLA alleles at two- and four-digit resolution using ethnicity-specific reference panels (see Methods). The first two digits specify a group of HLA alleles known as super-types as defined by older typing methodologies. The third through fourth digits specify nonsynonymous substitutions. Moreover, we imputed individual amino acid polymorphisms at class I (*HLA-A*, *-B*, and *-C*) and class II (*HLA-DQB1*, *-DQA1* and *-DRB1*) genes.

In East Asian cohorts, stepwise conditioning on classical HLA alleles defined two independent risk alleles, *DRB1*1501* (OR = 3.81, Wald test $P = 2.0 \times 10^{-49}$) and *DRB1*0301* (OR$_{conditioned}$ = 3.88, Wald test $P = 4.5 \times 10^{-24}$, Fig. 2c, Supplementary Table 3). In the analysis of polymorphic amino acid sites, genetic variation at only two codons encoding residues at positions 13 and 71 in DRβ, explained the entire *HLA-DRB1* signal (Fig. 3a, Supplementary Table 4). Specifically, DRβ position 13 occupied by Arginine (OR = 3.68, 95% CI: 2.74–4.95) or Serine (OR = 2.76, 95% CI: 2.06–3.71), and position 71 occupied by Lysine (OR = 3.10, 95% CI: 2.49–3.86) or Alanine (OR = 2.96, 95% CI: 2.55–3.45) conveyed the greatest risk (Supplementary Table 5). Consistent with a prior study in Chinese patients[9], these amino acids define the classical risk alleles *DRB1*1501* and *DRB1*0301*

(Supplementary Table 6), and their side chains map adjacent to each other within the antigen-binding pocket of the β-chain of DR (Fig. 3c).

The top Asian risk allele *DRB1*1501* had no significant risk effect in Europeans despite its frequency being comparable between populations (control freq. 10% vs. 8% in Europeans and East Asians, respectively). The most strongly associated European risk allele was *DQA1*0501* (OR = 2.88, Wald test $P = 5.7 \times 10^{-93}$, Fig. 2d). After conditioning the locus on *DQA1*0501*, *DRB1*0301* remained genome-wide significant (OR$_{conditioned}$ = 2.00, Wald test $P = 2.0 \times 10^{-19}$, Supplementary Table 7) suggesting that this risk allele is shared between Asian and European populations. We note that *DQA1*0501* allele is twice as common in Europeans compared to Asians (control freq. 30% vs. 14%). Moreover, *DQA1*0501* and *DRB1*0301* are in imperfect LD that is stronger in Europeans ($r^2 = 0.40$) compared to East Asians ($r^2 = 0.29$). Although a weak effect of *DQA1*0501* was apparent in our Asian cohorts (Fig. 2c, Supplementary Table 3), this allele became non-significant after conditioning on *DRB1*0301*. In contrast, *DQA1*0501* exhibited a genome-wide significant risk effect after conditioning on *DRB1*0301* in Europeans (OR$_{conditioned}$ = 2.40, Wald test $P = 1.8 \times 10^{-18}$).

Given that our HLA imputation reference panels were considerably smaller for East Asians compared to Europeans, we sought additional validation of the observed classical HLA associations that were Asian-specific. We therefore created another reference panel based on the MHC sequence data from Zhou et al.[10] including 10,689 control individuals of Han Chinese ancestry. Using SNP2HLA software, we then re-imputed classical HLA alleles for our East Asian cohorts. We used the same quality control filters (MAF > 0.01 and imputation $R^2 > 0.8$) and methods for association testing as described above. We observed no major differences in the association statistics for the two Asian risk alleles, *DRB1*1501* (OR = 3.49, $P = 3.85e-40$) and *DRB1*0301* (OR = 4.08, $P = 6.3E-24$), demonstrating that these effects do not represent artifacts of smaller imputation panels.

We next performed the analysis of HLA amino acid substitutions in Europeans. Consistent with the association analyses of classical alleles, five bi-allelic sites in DQA1 that correlate with the *DQA1*0501* allele were most strongly associated with the risk of MN (75Ser-107Ile-156Leu-161Glu-163Ser, Wald test $P = 5.7 \times 10^{-93}$, Fig. 3b, Supplementary Tables 8–10). Conditioning on this haplotype in Europeans uncovered a second independent signal in *HLA-DRB1*, position 74 (Supplementary Tables 8 and 9), with Arginine representing the key risk residue (OR 2.86, 95% CI: 2.54–3.23). This residue defines the European *DRB1*0301* risk haplotype (Supplementary Tables 9 and 10). Notably, positions 74 (Europeans) and 71 (East Asians) are separated by a single turn along the α-helix, and their side chains are spatially close to that of position 13, located on the beta-sheet floor with its side chain oriented into the peptide-binding groove (Fig. 3c).

To confirm ethnicity-specific HLA effects, we repeated stepwise conditioning in a joint stratified analysis of all cohorts using bi-allelic tests of HLA alleles with formal tests of heterogeneity (Supplementary Table 11, Fig. 4a). The top classical allele supported by all cohorts regardless of ethnicity was *DRB1*0301*

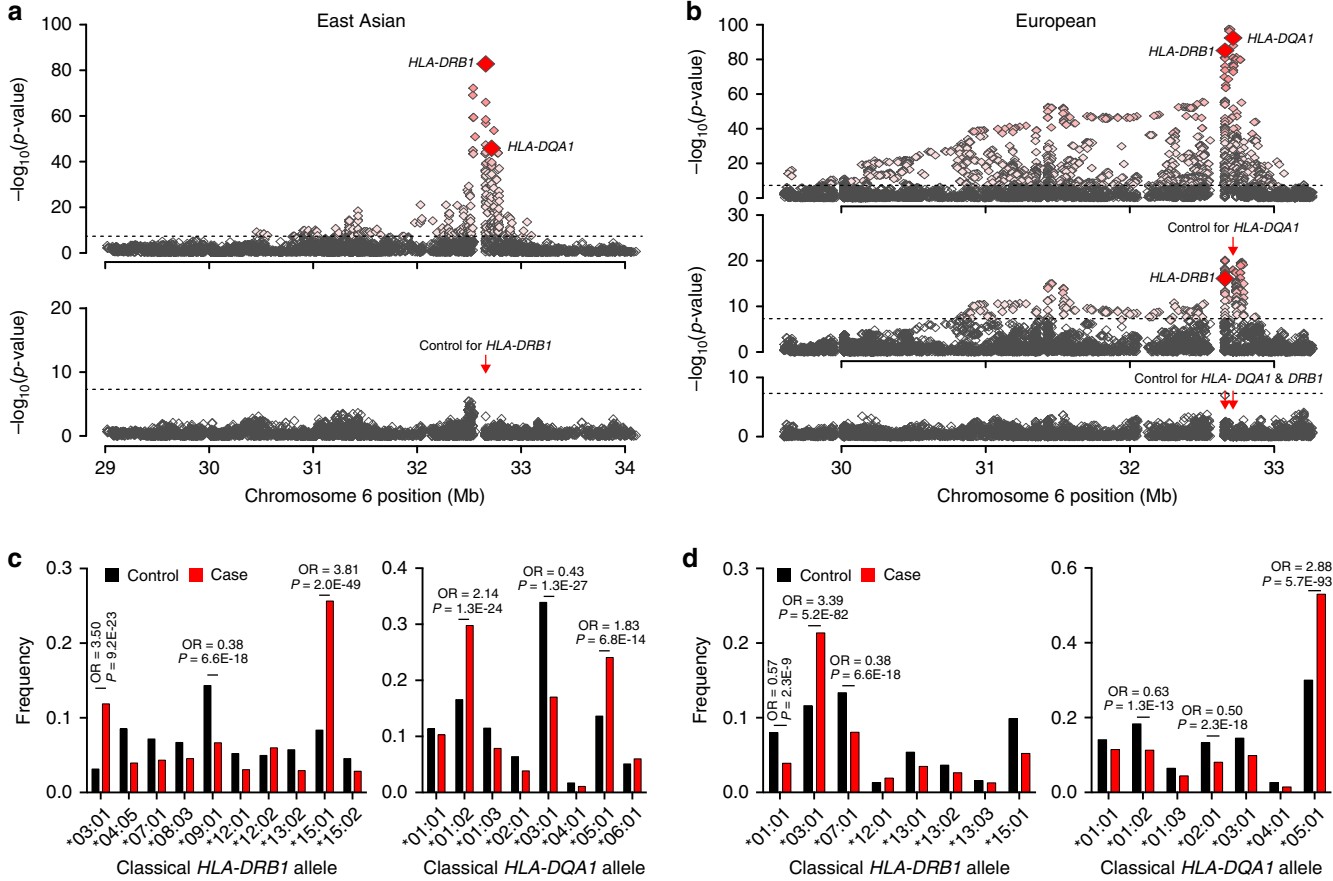

**Fig. 2 Ethnicity-specific association analyses point to *HLA-DRB1* and *HLA-DQA1* as the top associated genes. a** Regional plot for East Asian cohorts that includes association statistics for all imputed variants and classical HLA alleles; the strongest association signal was for *HLA-DRB1* gene (upper panel); the results for *HLA-DQA1* gene are highlighted for reference; after adjusting for all *DRB1* classical alleles (red arrow), there were no residual associations across the entire 3-Mb region (lower panel). The dotted horizontal line indicates the genome-wide significance threshold ($\alpha = 5 \times 10^{-8}$). **b** In Europeans, the strongest HLA gene association was for the *HLA-DQA1* gene (upper panel); after controlling for all classical *DQA1* alleles (red arrow), *HLA-DRB1* gene remained genome-wide significant (middle panel); after controlling for both *DRB1* and *DQA1*, there were no significant associations in the region (lower panel), suggesting that variation in both genes explains the entire signal. **c** East Asian frequency distributions of classical *DRB1* and *DQA1* alleles (four-digit resolution) for cases (red) and controls (black); unadjusted ORs and *P*-values provided for genome-wide significant alleles. **d** The European frequency distributions of classical *DRB1* and *DQA1* alleles (four-digit resolution) for cases (red) and controls (black); unadjusted ORs and *P*-values provided for genome-wide significant alleles.

(OR = 3.71, Wald test $P = 2.9 \times 10^{-127}$). After conditioning for *DRB1\*0301*, the top classical allele was *DQA1\*0501* (OR$_{\text{conditioned}}$ = 1.80, Wald test $P = 1.1 \times 10^{-30}$), but this association was supported predominantly by Europeans. After controlling for both *DRB1\*0301* and *DQA1\*0501*, the top allele was *DRB1\*1501* (OR$_{\text{conditioned}}$ = 1.94, Wald test $P = 4.7 \times 10^{-29}$), but the risk effect was supported exclusively by East Asians (heterogeneity $I^2 = 97.5$, Cochrane's Q-test $P < 0.05$).

***PLA2R1* locus and its genetic interactions**. Consistent with prior GWAS, the most significant non-HLA locus resided on chromosome 2q24.2[6]. The top SNP was in the first intron of *PLA2R1*, which encodes the main podocyte autoantigen in primary MN. This signal was supported by both ethnicities, but the effect appeared stronger in East Asians (OR = 2.81, Meta-analysis $P = 3.5 \times 10^{-61}$) compared to Europeans (OR = 1.98, Meta-analysis $P = 4.7 \times 10^{-48}$, Table 2). After conditioning the association on the top SNP, rs17831251, there was no residual association at this locus, suggesting a common risk haplotype in both ethnicities (Fig. 1b).

We next refined the previously reported genetic interactions between the *PLA2R1* locus and HLA risk haplotypes. The *PLA2R1* risk genotype exhibited significant multiplicative interaction with both Asian and European HLA risk haplotypes (Fig. 4b, c), with the risk homozygosity at both loci associated with 89-fold increased odds of disease risk in East Asians [OR = 88.8 for double risk homozygotes (N cases/controls = 103/10) vs. double protective homozygotes (N cases/controls = 15/152), 95% CI: 38.0–207.3, Interaction test $P = 7.8 \times 10^{-3}$] and 14-fold in Europeans [OR = 14.1 for double risk homozygotes (N cases/controls = 291/89) vs. double protective homozygotes (N cases/controls = 52/237), 95% CI: 10.0–22.1, Interaction test $P = 6.4 \times 10^{-5}$]. Because the effect modification was weaker in Europeans, we next repeated interaction testing separating individual HLA risk haplotypes (Fig. 4d, e). Notably, the interaction in Europeans was driven predominantly by the *DRB1\*0301-DQA1\*0501* haplotype [OR = 28.7 for double risk homozygotes (N cases/controls = 115/13) vs. double protective homozygotes (N cases/controls = 75/339), 95% CI: 15.1–54.4, Interaction test $P = 2.2 \times 10^{-3}$]. After removing its effect, *DQA1\*0501* had no residual interaction with *PLA2R1*

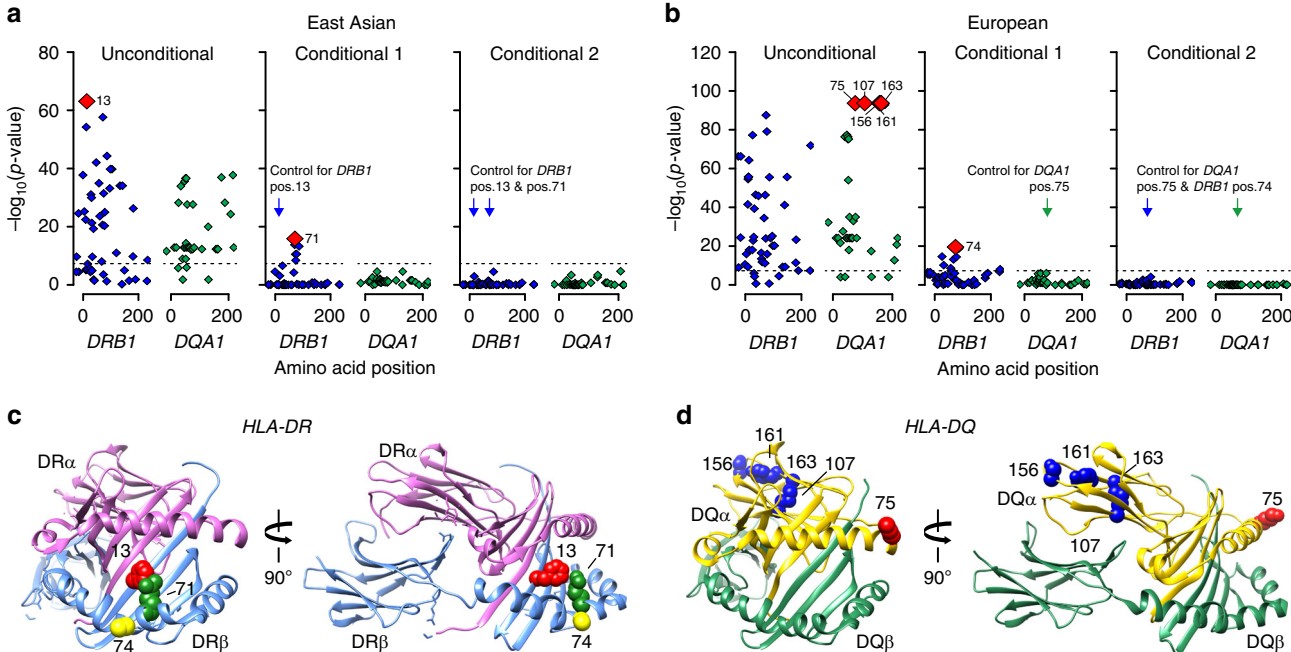

**Fig. 3 Ethnicity-specific association analyses of DRβ1 and DQα1 amino acid sequence. a** East Asian analysis of polymorphic amino acid positions within DRβ1 (blue) and DQα1 (green) molecules using conditional haplotype tests; the horizontal dash line marks the genome-wide significance level. The most strongly associated polymorphic site was position 13 in *DRB1* (left panel); after controlling for this position, position 71 in *DRB1* remained genome-wide significant (middle panel); after adjusting for both positions, there were no residual associations (right panel). **b** European analysis of polymorphic amino acid positions within DRβ1 (blue) and DQα1 (green); the DQA1 haplotype defined by amino acid positions 75, 107, 156, 161 and 163 (left panel) provided the strongest signal; after conditioning on this haplotype, position 74 in *DRB1* remained genome-wide significant (middle panel); no additional independent positions were found upon further conditioning (right panel). **c** Protein structure of the DR molecule including α chain (pink ribbon) and β chain (blue ribbon), the side chains of amino acids at DRβ1 positions 13, 71, and 74 are located adjacent to each other in the P4 pocket of the peptide-binding groove. **d** Protein structure of the DQ molecule including α chain (yellow ribbon) and β chain (green ribbon); all five bi-allelic amino acid sites in DQα1 are in perfect LD with each other and define the top associated *DQA1*0501* allele in Europeans.

(Interaction test $P = 0.1$). Similarly, there was no significant interaction between *DQA1*0501* allele and *PLA2R1* locus in East Asians. These analyses suggest that the *PLA2R1* locus interactions are driven primarily by the *DRB1* alleles.

We annotated all SNPs in LD with rs17831251 for potential impact on the structure and/or transcriptional regulation of *PLA2R1*. We found two common missense variants in moderate LD, rs35771982 (p.H300D, $r^2 = 0.69$) and rs3749117 (p.M292V, $r^2 = 0.68$), but the effects of these variants were considerably weaker compared to rs17831251, suggesting that they are unlikely to represent causal variants (Supplementary Table 12). Our tissue-specific functional scoring method for non-coding variants based on the ENCODE and Roadmap Epigenetics data[11] prioritized another variant in intron 1, rs17241973 ($r^2 = 0.93$ with rs17831251) that intersects a putative enhancer element across multiple tissues (Supplementary Fig. 6). Both rs17831251 and rs17241973 exhibit strong cis-eQTL effects on *PLA2R1* expression wherein the MN risk alleles associate with lower mRNA expression of *PLA2R1* across multiple tissues in GTEx[12], but this effect appears reversed for the kidney tissue (Supplementary Fig. 7). To further confirm these kidney-specific effects, we used gene expression data from manually micro-dissected human kidney compartments of 166 NEPTUNE participants[13]. We detected suggestive glomerular eQTL effects that were weak, but direction-consistent with GTEx for rs17831251 (Wald test $P = 0.055$) and rs17241973 (Wald test $P = 0.024$), wherein MN risk allele were associated with increased glomerular *PLA2R1* mRNA levels (Supplementary Fig. 8).

Because kidney tissue compartments are not well represented in either ENCODE or Roadmap datasets, we next examined the

genomic location of rs17831251 and rs17241973 in relationship to the recently published kidney compartment-specific chromatin landscape[14] (Supplementary Fig. 9); rs17241973 lies within intron 1 of *PLA2R1* in open chromatin that is active in both glomerular and tubular compartments and is contiguous with the gene promoter. In contrast, rs17831251 lies within a broad region of increased chromatin accessibility in glomeruli, and is only 2.1-kb away from a glomerulus-specific DHS that contains a high-confidence NFKB1-binding motif. In glomerulus-specific chromatin conformation (Hi-C) data, both rs17831251 and rs17241973 make regional and distal contacts with other glomerular DHS, emphasizing a composite cis-regulatory module for *PLA2R1* gene expression.

**Novel loci encoding *NFKB1* and *IRF4*.** The 4q24 locus contains the *NFKB1* gene, which encodes an active DNA binding subunit of the NF-κB transcriptional complex. The top SNP, rs230540 (OR = 1.25, Meta-analysis $P = 3.4 \times 10^{-12}$), is an intronic variant predicted to have a functional effect specific to immune cells (Supplementary Fig. 10). In agreement with our prediction, rs230540 has been associated with higher mRNA expression of *NFKB1* in whole blood ($P = 2.6 \times 10^{-11}$)[15] and in CD4+ T cells ($P = 2.0 \times 10^{-9}$)[16]. Consistent with pro-inflammatory effects of NF-κB, the MN risk haplotype at this locus determined higher leukocyte counts[17] and increased risk of ulcerative colitis[18,19] and primary biliary cholangitis[20,21] (Supplementary Table 13, Fig. 5a). *NFKB1* is also expressed in human podocytes[22], as well as primary human glomerular and tubular epithelial cell cultures[14], and rs230540 intersects an active glomerular DHS with several

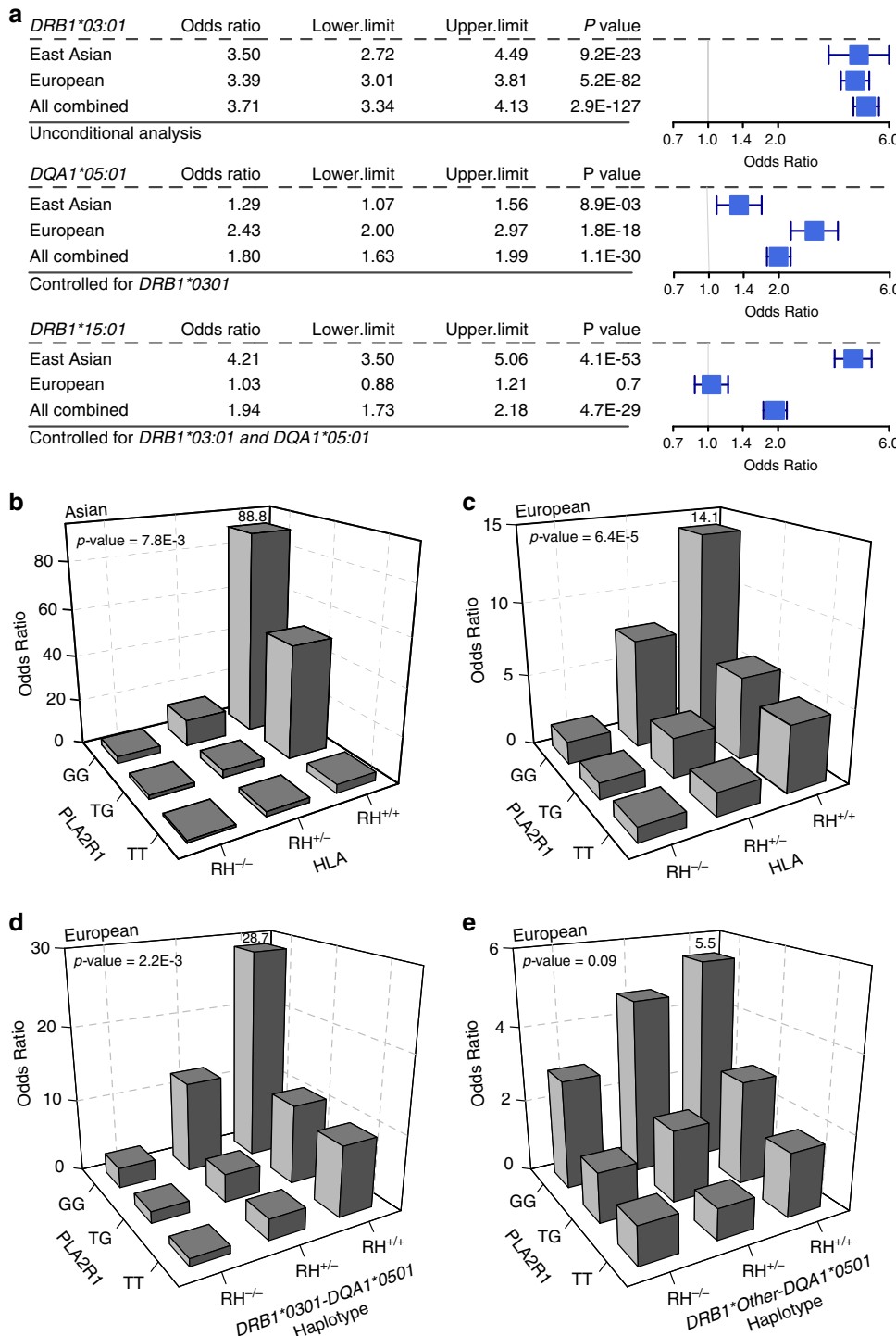

**Fig. 4 Ethnicity-specific HLA allelic effects and genetic interactions. a** Stepwise conditioning of HLA risk alleles by ethnicity; Top: Unconditioned effect estimates, 95% confidence intervals, and *P*-values for *DRB1*0301* demonstrate similar effects in East Asian and European cohorts. Middle: Effect estimates for *DQA1*0501* after controlling for *DRB1*0301* demonstrate significant heterogeneity, with stronger effects in the European cohorts (Cochrane's Q *P*-value < 0.05). Bottom: Effect estimates for *DRB1*1501* after controlling for both *DRB1*0301* and *DQA1*0501* demonstrate significant heterogeneity with risk effect in East Asians but no effect in Europeans (Cochrane's Q *P*-value < 0.05). **b** PLA2R1 genotype interaction with East Asian HLA risk haplotypes, *DRB1*0301* or *DRB1*1501* (N cases/controls = 803/1,956, $P = 7.8 \times 10^{-3}$). **c** PLA2R1 genotype interaction with European HLA risk haplotypes, *DRB1*0301* or *DQA1*0501* (N cases/controls = 1880/2627, multiplicative interaction test $P = 6.4 \times 10^{-5}$). **d** PLA2R1 genotype interaction with *DRB1*0301-DQA1*0501* risk haplotype in Europeans (multiplicative interaction test $P = 2.2 \times 10^{-3}$). **e** No significant interaction between PLA2R1 and DQA1 risk haplotypes other than *DQA1*0501-DRB1*0301* in Europeans (multiplicative interaction test $P = 0.09$). RH risk haplotype.

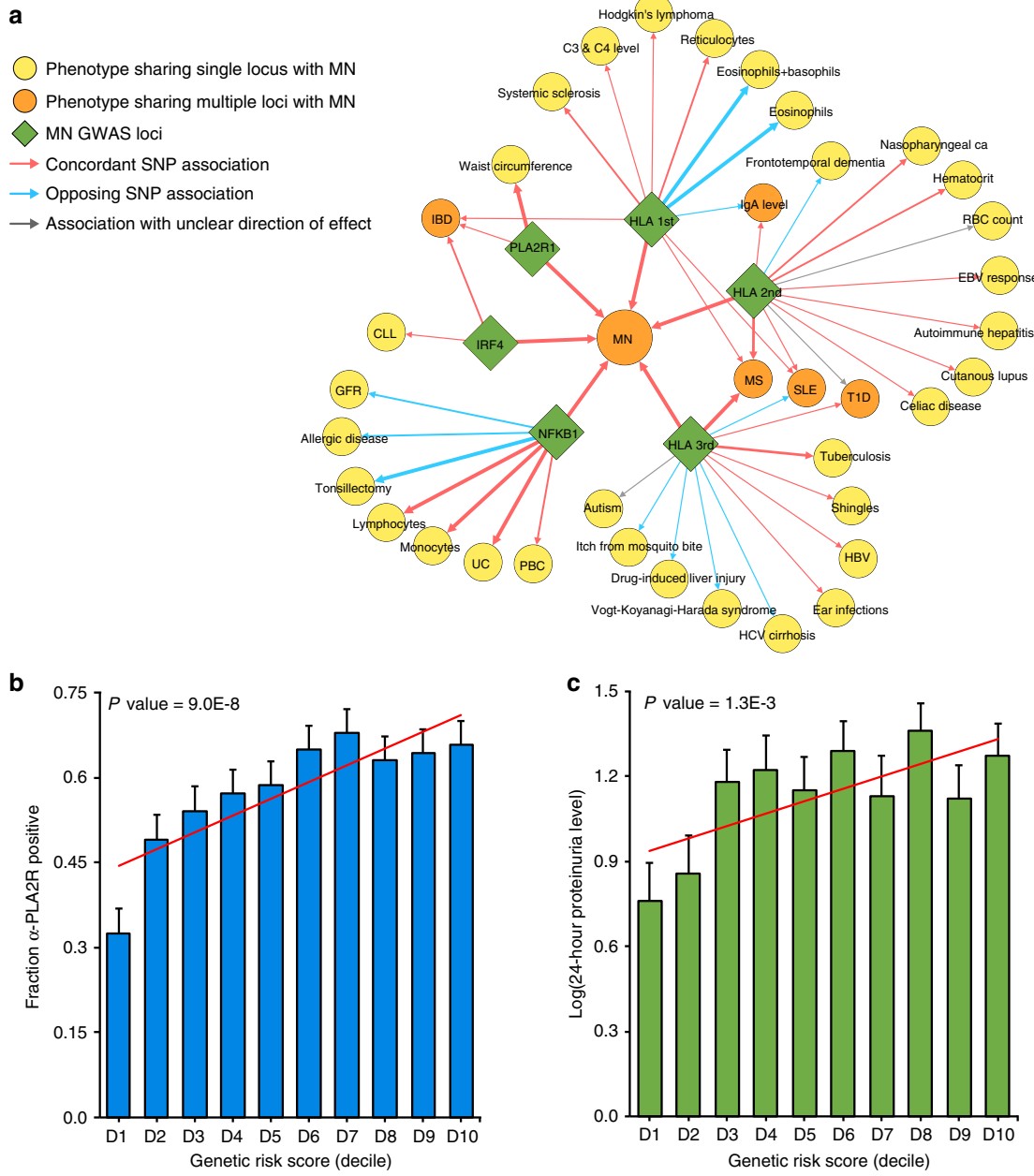

**Fig. 5 Pleiotropic effects of the MN loci and their clinical correlations. a** The pleiotropy map was constructed based on overlapping genome-wide significant loci reported in the GWAS Catalogue: traits sharing a single locus with MN are indicated in yellow; traits sharing multiple loci are indicated in orange; arrows represent allelic associations that are identical to, or in tight LD ($r^2 > 0.8$) with the MN risk alleles; arrow thickness is proportional to $r^2$ between alleles; concordant effects are indicated in red and opposed effects in blue. IBD: inflammatory bowel disease, includes ulcerative colitis (UC) and Crohn's disease (CD); PBC primary biliary sclerosis, GFR glomerular filtration rate, CLL chronic lymphocytic leukemia, HBV hepatitis B virus; **b** Significant positive correlation of the GRS with anti-PLA2R antibody seropositivity ($N = 1114$ cases, Wald test $P = 9.0 \times 10^{-8}$) and **c** log-transformed 24-h proteinuria at diagnosis ($N = 1329$ cases, Slope test $P = 1.3 \times 10^{-3}$). The x-axis depicts deciles of GRS; error bars correspond to standard errors; the P-values are adjusted for age, sex, and ethnicity.

glomerular Hi-C contact sites[14] (Supplementary Fig. 11). Notably, the MN risk allele has previously been associated with lower estimated glomerular filtration rate in GWAS of renal function[23,24], thus this locus may be more broadly associated with the risk of kidney disease.

The top SNP on chromosome 6p25.3, rs9405192 (OR = 1.29, Meta-analysis $P = 1.4 \times 10^{-14}$), resides upstream of *IRF4* gene, which belongs to the family of transcription factors regulating interferon-inducible genes. IRF4 is lymphocyte specific and negatively regulates Toll-like-receptor signalling that is central to the activation of innate immune system; this gene is known to

be under the transcriptional control of the NF-kB complex[25–27]. Unlike *PLA2R1* and *NFKB1*, *IRF4* does not appear to be expressed in human kidney cells by single nuclei RNA-seq[28] (Supplementary Fig. 12). We did not find functional or coding SNPs in LD with rs9405192, nor did we observe any cis-eQTL effects for this variant, thus the precise mechanism underlying this association remains unknown. However, the analysis of binding sites for individual components of the NF-κB complex in lymphocytes[29] suggested binding of the complex in close proximity of rs9405192 (Supplementary Fig. 13). The risk allele at this locus is also in strong LD with variants previously

associated with increased risk of inflammatory bowel disease[19], and in weaker LD with several risk variants for chronic lymphocytic leukemia (Supplementary Table 13, Fig. 5a), suggesting the pattern of pleiotropy that is similar to the *NFKB1* locus. Nevertheless, we detected no statistically significant genetic interactions of *IRF4* and *NFKB1* loci.

In addition, we systematically annotated all other suggestive non-HLA loci defined by $P < 5.0 \times 10^{-5}$ and these results are summarized in Supplementary Table 14. To enhance potential genetic discovery of novel podocyte antigens, we also repeated genome scans after conditioning for the *PLA2R1* locus, but detected no additional suggestive loci.

**SNP-based heritability and risk explained by GWAS.** Using our genotype data and genome-based restricted maximum likelihood method (GREML)[30], we estimated the overall SNP-based heritability of MN at 0.43 (SE = 0.039) in East Asians and 0.36 (SE = 0.0046) in Europeans. Remarkably, all genome-wide significant risk alleles exhibited unusually large effect sizes for GWAS. In order to quantify the fraction of disease variance cumulatively explained by genome-wide significant SNPs and their interactions, we performed ethnicity-specific GRS analyses (see Methods). Each GRS was expressed as a weighted sum of risk alleles with weights defined by their mutually adjusted effect estimates and included the 3 independent non-HLA SNPs (rs6707458, rs230540, rs9405192) as well as ethnicity-specific HLA risk alleles and their interactions. This included rs9269027, rs1974461, and rs9269027*rs6707458 interaction term for East Asians, and rs9271541, rs9265949, rs2858309 and rs9271541*rs6707458 interaction term for Europeans (Supplementary Table 15). The GRS calculated using this method explained 32% disease risk in East Asians, 25% in Europeans, and 29% of overall disease risk across all cohorts combined. Remarkably, the magnitude of the GRS effect was comparable to rare, highly penetrant mutations causing Mendelian forms of kidney disease, with individuals in the top decile of GRS having 30 to 40-fold higher disease risk compared to the lowest decile (Fig. 6a, b).

**Clinical correlations of the GRS.** For a subset of patients with available clinical data, we performed genetic correlation analyses with selected clinical features reflective of disease severity. The GRS was positively correlated with PLA2R antibody seropositivity (Wald test $P = 9.0 \times 10^{-8}$), and in those with detectable antibodies, higher titers at the time of biopsy (Slope test $P = 1.2 \times 10^{-9}$, Fig. 5b). The GRS also predicted worse proteinuria at the time of biopsy, which represents the key marker of MN severity and prognosis (Slope test $P = 1.3 \times 10^{-3}$, Fig. 5c). Other clinical features, such as age at diagnosis, renal function, or serum albumin levels at the time of biopsy were not significantly correlated with the GRS after multivariate adjustment (Supplementary Table 16).

**Potential diagnostic implications of the GRS.** Although the diagnosis of MN is traditionally established by a kidney biopsy, the detection of circulating PLA2R antibodies by ELISA has recently emerged as a useful diagnostic modality[31]. In this study, we performed ELISA in sera obtained within 6 months of a diagnostic kidney biopsy in a total of 2331 individuals (1488 cases, 300 healthy controls, and 543 disease controls). In East Asians, we estimated that the standard ELISA cut-off of 20 U/mL provided 100% specificity and 60% sensitivity for the diagnosis of MN. In the analysis of Europeans, depending on the specific cohort, the same cut-off provided 99–100% specificity and 51–57% sensitivity (Supplementary Table 17). While the antibody level of 20 U/mL represents the manufacturer's recommended

cut-off, levels 2–20 U/mL are frequently considered as borderline-negative, and levels <2 U/mL as negative[31]. In our cohorts, the cut-off 2 U/mL had inadequate diagnostic specificity (range 73–92%). These results confirm the key limitation of the PLA2R antibody ELISA, which has high specificity (99–100%) but low sensitivity (51–60%) at the standard recommended cut-off point; while lowering the cut-off increases sensitivity, it results in inadequate specificity. Consequently, the levels in the borderline-negative range (2–20 U/mL) are difficult to interpret clinically.

Given this limitation, we evaluated if the addition of genetic risk information can improve the performance of ELISA, especially in cases that fall in the borderline-negative range. First, we evaluated diagnostic properties of the GRS alone in our discovery cohorts. In East Asians, the genetic test had area under the receiver operating characteristics curve (AUROC) of 0.80 (95% CI: 0.78–0.82), while in Europeans the AUROC was 0.75 (95% CI: 0.74–0.77). Combining genetic and serologic tests in the form of a CRS provided superior case discrimination with AUROCs of 0.96 (95% CI: 0.95–0.98) in East Asians and 0.89 (95% CI: 0.87–0.91) in Europeans (Fig. 6, Supplementary Table 18).

We next tested the GRS performance in several external validation cohorts, including three independent GWAS cohorts of European ancestry as well as in the European-American NEPTUNE participants with incident nephrotic syndrome (Supplementary Table 18C-E). Overall, the effects and the diagnostic performance of the GRS were comparable between the European discovery and each validation cohort (Fig. 6c). When combined with the serum antibody titer, the CRS achieved AUROC of 0.96 (95% CI: 0.94–0.97) across all validation cohorts combined (Supplementary Table 18E, Fig. 6f).

In the subgroup analyses, we compared the diagnostic properties of GRS and CRS by the antibody status in all cohorts pooled by ancestry (Fig. 7). These analyses demonstrated that both GRS and CRS were predictive of case status even for antibody-negative MN. Importantly, the CRS continued to have excellent performance in classifying the borderline-negative cases (antibody level 2–20 U/mL range), with AUROCs of 0.98 (95% CI: 0.97–0.99) in East Asians and 0.95 (95% CI: 0.93–0.96) in Europeans. Notably, among all cases for which the ELISA test was either negative or inconclusive, adding genetic information in the form of CRS can establish the diagnosis in 20–37% cases with 99% specificity. The comparison of AUROCs between GRS, CRS, and serum anti-PLA2R Ab test by ancestry is provided in Supplementary Fig. 14, and the clinical implications of these findings are summarized in Supplementary Note 1 and Supplementary Table 19.

Lastly, we expanded our validation studies to non-European participants of the NEPTUNE study (Supplementary Tables 20 and 21). Although the European risk score performance was diminished in Hispanic Americans, the European GRS performed well in African Americans, and this is despite substantial differences in risk allele frequencies between Europeans and Africans (Supplementary Table 22). Similar to the European validation cohorts, the European GRS was superior compared to the trans-ethnic GRS when applied to the NEPTUNE minority populations, while the Asian GRS had relatively poor performance in both African-American and Hispanic/Latino cohorts (Supplementary Table 21, Supplementary Fig. 15).

## Discussion

Our study provides important insights into an autoimmune disease and the genetic architecture of MN. First, we discover novel genome-wide significant risk loci for MN with large effects encoding two transcriptional master regulators of inflammation, NFKB1 and IRF4. The association at the *NFKB1* locus highlights

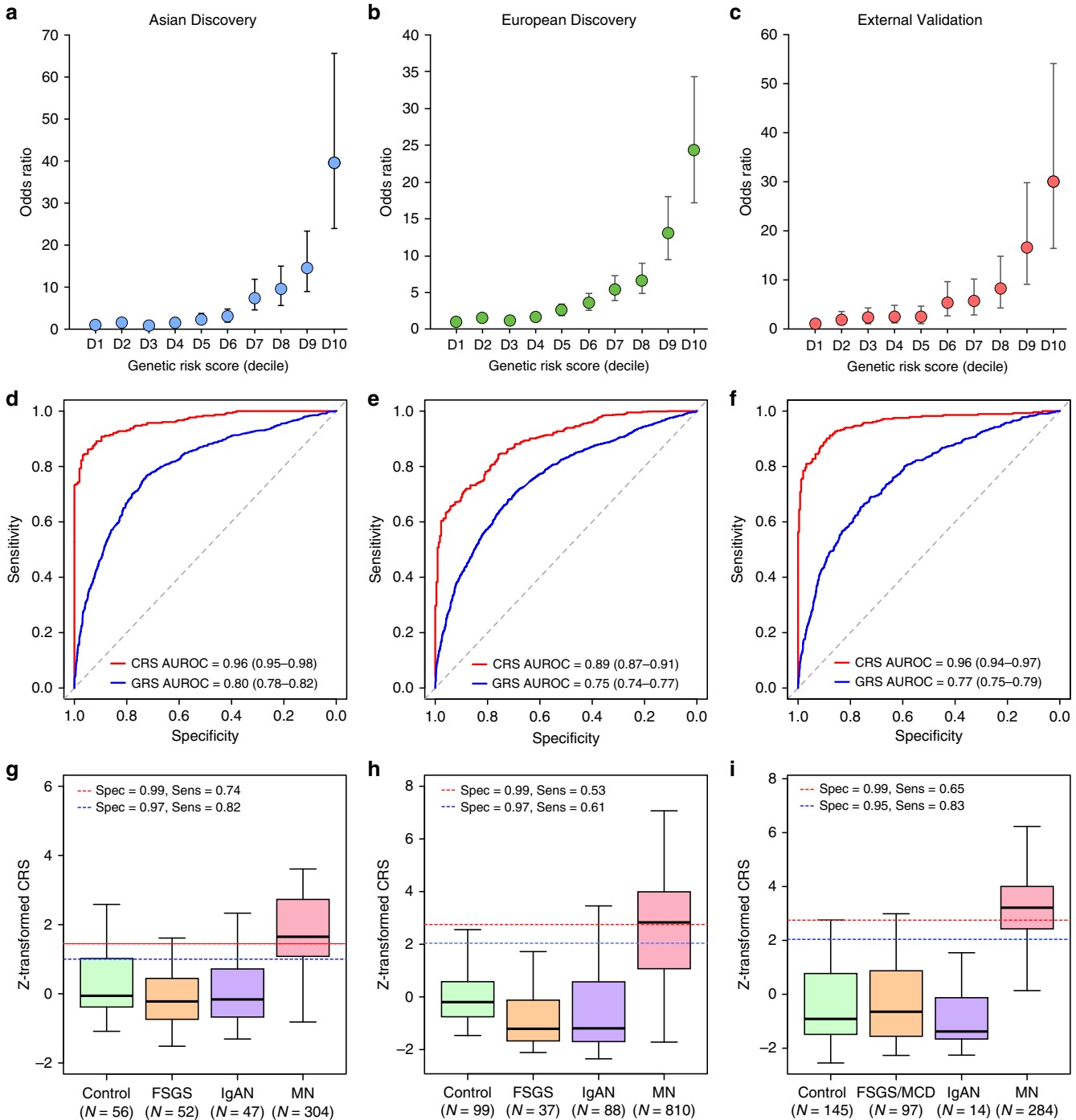

**Fig. 6 Diagnostic performance of the genetic risk score (GRS) and the combined risk score (CRS).** Genetic effects expressed as odds ratios (OR) and 95% confidence intervals in reference to the lowest decile of the GRS distribution for **a** East Asian discovery, **b** European discovery, and **c** European validation cohorts combined. GRS and CRS Receiver Operating Characteristics (ROC) curves for **d** East Asian discovery, **e** European discovery, and **f** European validation cohorts combined; AUROC Area under the ROC curve. Distributions of the CRS between healthy controls, diseased controls, and MN cases for **g** East Asian discovery, **h** European discovery, and **i** European validation cohorts combined. The box plots depict medians (horizontal lines), interquartile ranges (boxes), and minimum/maximum values (whiskers). The discovery CRS cut-offs of 1.00 and 1.45 in East Asians and 2.05 and 2.72 in Europeans have 97% and 99% specificity, respectively. The same European cut-offs were applied to the validation cohorts for comparisons of specificity and sensitivity.

the role of the canonical NF-κB pathway in primary MN. Upon activation by pro-inflammatory signals, NFKB1 undergoes proteasome processing to p50, an active DNA binding subunit of the NF-κB complex. Inappropriate activation of this pathway has previously been studied in progressive diabetic nephropathy[32],

and in the context of inflammatory diseases, including IBD[33,34] and MN[35–37]. Importantly, the MN risk allele at this locus has a concordant effect on the risk of ulcerative colitis[18,19] and primary biliary cirrhosis[20,21]. It has also been associated with increased mRNA expression of *NFKB1* in cis, and reduced DNA

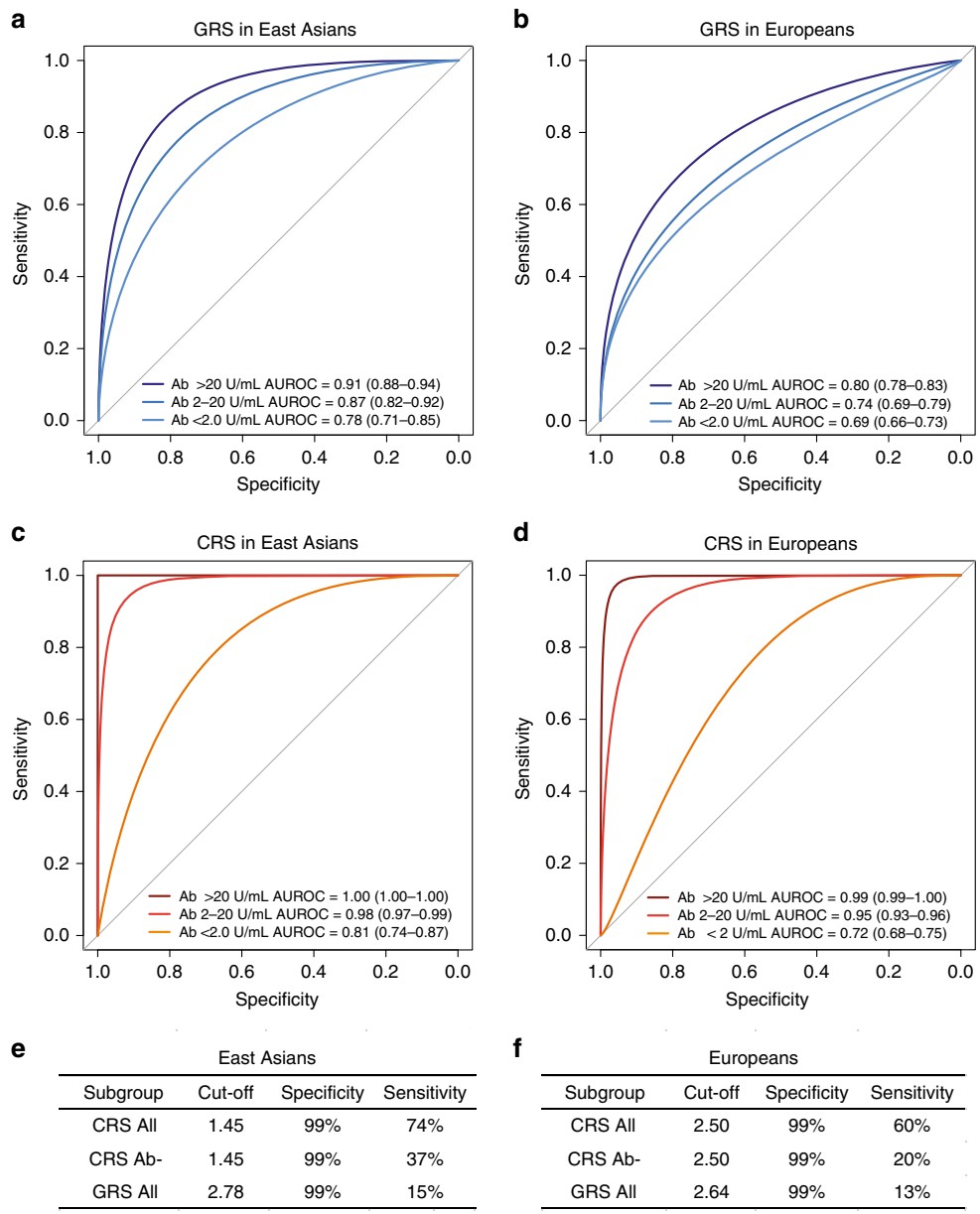

| East Asians | | | |
|---|---|---|---|
| Subgroup | Cut-off | Specificity | Sensitivity |
| CRS All | 1.45 | 99% | 74% |
| CRS Ab- | 1.45 | 99% | 37% |
| GRS All | 2.78 | 99% | 15% |

| Europeans | | | |
|---|---|---|---|
| Subgroup | Cut-off | Specificity | Sensitivity |
| CRS All | 2.50 | 99% | 60% |
| CRS Ab- | 2.50 | 99% | 20% |
| GRS All | 2.64 | 99% | 13% |

**Fig. 7 Diagnostic properties of the genetic risk score (GRS) and combined risk score (CRS) stratified by anti-PLA2R antibody status.** Comparisons of receiver operating characteristic (ROC) curves to discriminate antibody positive (PLA2R Ab > 20 U/mL), borderline negative (PLA2R Ab 2–20 U/mL), and negative (PLA2R Ab < 2 U/mL) cases of primary MN from all available healthy and diseased controls combined for **a** GRS in East Asian discovery cohorts, **b** GRS in European discovery and validation cohorts, **c** CRS in East Asian discovery cohorts, **d** CRS in European discovery and validation cohorts. Overall sensitivities for risk score cut-offs corresponding to 99% specificities in (**e**) East Asian discovery cohorts and (**f**) European discovery and validation cohorts; CRS All: all patients with serum Ab measurements by ELISA within 6 months of a diagnostic kidney biopsy; CRS Ab-: patient subgroup with PLA2R Ab < 20 U/mL, and GRS All: all patients with GWAS data available for GRS calculation. AUROC: area under the ROC curve (95% confidence interval).

methylation in trans across >400 CpGs that overlap with NF-κB-binding sites, suggesting enhanced baseline activity of NF-κB[38]. The NF-κB complex is known to up-regulate IRF4 expression with cross-regulatory feedback loops between *NFKB1* and *IRF4* described in several prior studies[25–27]. Taken together, NFKB1 and IRF4 loci participate in a common regulatory pathway in immune cells, and our genetic findings clearly establish a critical role of this pathway in the pathogenesis of MN.

Second, due to the bi-ethnic composition of our cohorts, we were able to refine ethnicity-specific effects at the *HLA* locus, defining *DRB1*1501* as a major risk allele in East Asians, *DQA1*0501* in Europeans, and *DRB1*0301* in both ethnicities.

These findings suggest that different epitopes are likely presented to T cells to initiate the anti-PLA2R response in East Asians and Europeans. We also identified specific high-risk amino acid substitutions, at positions 13, 71, and 74, mapping to the P4 pocket of DRβ1. Although the same positions contribute to the risk of T1D[39] and rheumatoid arthritis[40], the effects of individual residues at each position are discordant, likely reflecting differences in target epitopes.

Third, we confirm that a single haplotype at the *PLA2R1* locus conveys the disease risk in both East Asians and Europeans, and exhibits genetic interactions with *HLA-DRB1* risk alleles. Our analysis supports a regulatory function of the *PLA2R1* risk

haplotype. The candidate causal variant resides in the first intron of *PLA2R1* and intersects a predicted enhancer element. While this variant is normally associated with suppressed *PLA2R1* transcription across multiple tissues, it appears to increase expression of *PLA2R1* in the kidney. This finding highlights the importance of studying target tissues and is consistent with the findings that among CKD loci that are transcriptionally active in renal tissue, 15.8% of effects are kidney-specific[41]. Notably, the top variants at the *PLA2R1* locus also intersect a putative NF-κB binding site in lymphocytes, although no similar data is presently available for podocytes. Further experimental work is thus needed to test if the glomerular-specific eQTL effect is under the transcriptional control of NF-κB. Moreover, larger glomerular compartment-specific datasets will be needed to confirm the observed eQTL effects.

Another observation is that all four genome-wide significant risk loci (*PLA2R1, IRF4, NFKB1,* and *HLA*) exhibit highly pleiotropic effects and all four lead SNPs have a concordant effect on the risk of inflammatory bowel disease (IBD). This observation suggests shared pathogenic mechanism between IBD and MN. Considering that MN is an orphan disease without a targeted treatment, there may now be opportunities for drug repositioning approaches from IBD, where several new anti-inflammatory agents are currently under development. Our study suggests that the NF-κB and interferon pathways may represent particularly attractive drug targets.

Remarkably, our GWAS loci are highly predictive of the disease status and jointly explain up to one third of disease risk, an exceptionally large fraction for common alleles. This may be partially explained by the fact that MN frequently occurs after the peak reproductive age, allowing the risk alleles to escape purifying selection. Moreover, even though the risk alleles are common, our interaction analysis demonstrates that specific high-risk genotype combinations are relatively rare in the general population, potentially explaining the low overall prevalence of MN[42]. The alternative hypothesis is that of balancing selection. NF-kB and IRF4 are both involved in immune defenses against common pathogens and some Phospholipase A2 ligands for PLA2R1 represent downstream NF-kB targets with antibacterial properties[43,44]. Therefore, the observed high frequencies of MN risk alleles could be explained by their protective effects against common infections.

Finally, a simple GRS based on our GWAS loci has excellent discriminant properties when combines with anti-PLA2R ELISA test. Importantly, the combined genetic-serum test has superior diagnostic properties compared to serologic test alone, mitigating the key issue of low sensitivity. The GRS provides complementary information to the serum test and correctly reclassifies 20–37% of antibody-negative cases, potentially sparing the need for a kidney biopsy in this large subgroup of patients. In the clinical settings where neither a serum test nor a kidney biopsy is possible, the GRS itself can establish a diagnosis of MN with 99% specificity in 13–15% of cases. The practical advantage of this approach is that the GRS can be readily determined at any time after birth and, unlike the serum test, it does not fluctuate with time or in relationship to the disease onset, activity, or treatment. One important limitation, however, is that genetic effects may be population-specific and may not be generalizable to populations not represented in our GWAS. The performance of the GRS is remarkably consistent in our discovery and validation cohorts, including African-Americans, but it appears to be lower in self-reported Latino/ Hispanics. Therefore, future efforts extending GWAS for MN to more diverse populations will be important.

In summary, we described a highly unusual genetic architecture of MN, including large effect sizes for a small number of common alleles and a strong evidence for ethnicity-specific

genetic interactions. These insights enabled formulation a powerful genetic disease predictor that provides means to enhance a non-invasive diagnosis of MN, and can be especially useful in the settings where kidney biopsy represents too great of a risk or is not readily available.

## Methods

**Study design overview**. We performed a genome-wide meta-analysis of eight discovery cohorts of East Asian and European ancestry (2956 cases and 7799 controls), all genotyped with high resolution arrays and imputed to ~7 million common high-quality markers using ancestry-matched reference panels. The top signals from the meta-analysis ($P < 5 \times 10^{-5}$) were typed in the additional East Asian replication cohort of 826 cases and 1239 controls. Subsequently, all cohorts (3782 cases and 9038 controls) were analyzed jointly to define genome-wide significant signals. All subjects provided informed consent to participate in genetic studies, and the Institutional Review Board of Columbia University as well as local ethics review committees for each of the individual cohorts approved our study protocol. The individual cohorts, genotyping methods, and quality control analyses are described in the Supplementary Methods.

**Primary association analyses and genome-wide meta-analyses**. Within each cohort, primary association scans were performed for markers that were common (MAF > 0.01) and imputed at high quality ($r^2 > 0.8$) using logistic regression under additive coding of dosage genotypes, and with adjustment for cohort-specific significant principal components (PCs) of ancestry. To quantify potential inflation of type I error due to stratification or technical artifacts, we estimated genomic inflation factors[45] for each genome-wide scan after excluding *HLA* and *PLA2R* loci. No substantial inflation was noted in any individual scan (lambda consistently <1.05 for each individual cohort). Subsequently, a fixed effects meta-analysis was performed to combine the results of the eight discovery cohorts using METAL[46]. Genome-wide distributions of *P*-values were examined visually using quantile-quantile plots for each individual cohort as well as for the combined analysis. The final meta-analysis quantile-quantile plot showed no global departures from the expected null distribution (Supplementary Fig. 3), with the genomic inflation factor estimated at 1.03 for the overall meta-analysis. Suggestive signals were defined by *P*-value < $5.0 \times 10^{-5}$. To declare genome-wide significance of a novel locus, we used the generally accepted *P*-value threshold of $5.0 \times 10^{-8}$.

**Conditional analyses**. To detect additional independent SNPs at genome-wide significant loci, we performed stepwise conditional analyses of each locus using logistic regression. This was done by including the genotype of conditioning SNP (s) under additive coding as covariate(s) in the outcome model. The conditional analyses were performed individually within each cohort and with adjustments for cohort-specific ancestry PCs. Subsequently, the conditioned summary statistics were combined across cohorts using fixed effects meta-analysis, similar to our primary association analyses.

**Credible set analyses**. For each of the three genome-wide significant non-HLA loci, we derived 99% credible sets using the trans-ethnic and ethnicity-specific meta-analysis results. First, we derived approximate Bayes factors from GWAS association statistics using Wakefield's formula, as implemented in the R package gtx[47]. Using CAVIAR software[48], we next calculated the posterior probability (PP) for each SNP driving the association signal at each locus. We assumed there was only a single causal variant at each locus, since no additional independent SNPs were detected on stepwise conditioning analyses. We derived both trans-ethnic as well as ethnicity-specific 99% credible sets based on ranking the variants by their PPs and adding the variants to the set until cumulative PP > 99% was reached for each region. The overlaps between ethnicity-specific and trans-ethnic analyses were visualized in Supplementary Fig. 5.

**HLA imputation**. Six discovery cohorts (Chinese, South Korean, Japanese, European-1, European-2, and Turkish) had primary genotype data available for HLA imputation and association testing. For each of these cohorts, we imputed classical HLA alleles at two- and four-digit resolution, as well as individual amino acid polymorphisms at class I (*HLA-A*, *-B*, and *-C*) and class II (*HLA-DQB1*, *HLA-DQA1* and *HLA-DRB1*) loci using SNP2HLAsoftware[49]. The European cohorts and the East Asian cohorts were imputed separately, using ethnicity-specific reference panels. For European reference, we used the pre-phased HLA reference dataset generated by the Type 1 Diabetes Genetics Consortium (T1DGC, 5,225 individuals)[49]. For our East Asian cohorts, we used the Pan-Asian HLA Reference Panel (268 individuals)[50]. For validation of classical HLA association results in East Asians, we built additional East Asian reference panel based on the MHC sequence data from Zhou et al. (10,689 Han Chinese)[10]. In the association analyses, we included only common HLA alleles (MAF > 0.01) that were imputed with high certainty ($R^2 > 0.8$).

**Statistical framework for HLA association testing.** Given that the frequency of HLA alleles can vary by ethnicity, we performed HLA association testing Europeans and East Asians separately. We used logistic regression models to test the additive effects of HLA allele dosages with adjustment for significant PCs of ancestry. For multi-allelic loci, we used the following logistic regression model:

$$\log(odds_i) = \beta_0 + \sum_{j=1}^{m-1} \beta_j x_{j,i} + \sum_{k=1}^{n} \beta_k P_{k,i} \quad (1)$$

where m indicates a total number of alleles at a specific multi-allelic locus, $j$ indicates a specific allele being tested, and $x_{j,i}$ is the imputed dosage for allele $j$ for individual $i$; $\beta_0$ represents the intercept and $\beta_j$ represents the additive effect of an allele $j$; $P_{k,i}$ denotes the value for $k$th PC of individual $i$, $n$ is the total number of significant PCs in the dataset, $\beta_k$ is the effect size of principal component $k$. We compared log-likelihoods of two nested models: the full model containing the test locus and relevant covariates with the reduced model without the test locus, but with the same set of covariates. The deviance was defined as $-2\times$log likelihood ratio, which follows a $X^2$-distribution with $m-1$ degrees of freedom, from which we calculated $P$-values. In addition to multi-allelic tests, we also performed bi-allelic tests of association for all individual SNPs, classical HLA alleles, and individual amino acid residues in HLA molecules. All analyses were performed using dosage method under additive coding. Stepwise conditioning analyses across the HLA region were performed using both multi-allelic and bi-allelic coding of HLA variants. In each round of stepwise conditioning, we first included the most significant variant as the covariate in the logistic regression model. If additional independently associated markers are detected, they are included as covariates in our subsequent models. We repeated these analyses until no residual associations across the entire locus were observed.

**Analysis of polymorphic amino-acid sites in HLA genes.** To test the effects of individual amino acid substitution sites within the *HLA-DRB1* and *HLA-DQA1* genes, we applied a conditional haplotype analysis using fully phased haplotypes across the HLA region. We tested each single amino acid position by first identifying the m possible amino acid residues occurring at that position and then using $m-1$ degrees of freedom test to derive $P$-values, with a single amino acid residue arbitrarily selected as a reference. For conditioning on individual amino acid sites, we used the following procedure: by adding a new amino acid position to the model, a total of $k$ additional unique haplotypes were generated and tested over the null model (without a new amino acid site) using the likelihood ratio test with $k$ degrees of freedom. If the new position was independently significant, we further updated the null model to include all unique haplotypes created by all amino acid residues at both positons to identify another independent position. The procedure was repeated until no statistically significant positions were observed.

**Testing for pairwise epistasis.** Multiplicative interactive effects were tested using logistic regression model; SNPs were coded under additive genotype coding (0, 1, 2), and interaction terms were defined as simple products of genotypes. To screen for interactions, we tested a lead SNP at each of the three non-HLA loci against each of the five independent HLA SNPs (three in Europeans and two in East Asians) resulting in a total of 15 independent tests. We additionally tested for all pairwise interactions between the three non-HLA loci shared between both ethnicities resulting in three additional tests. In each case, we used a likelihood ratio test comparing two nested models: the full model with both main effects and the interaction term to the reduced model with main effects only. Given a total of 18 pairwise interaction tests, we used a Bonferroni-corrected significance threshold of $0.05/18 = 2.8 \times 10^{-3}$. In secondary analyses, we explored if significant HLA risk haplotypes interact with the *PLA2R1* risk allele in the six cohorts with fully imputed classical HLA alleles. This included a total of 2759 East Asians (803 cases and 1956 controls) and 4507 Europeans (1880 cases and 2627 controls).

**Functional annotations of GWAS loci.** We used several different approaches to perform functional annotations of our significant loci. We first defined the region of each locus as $+/-400$ kb of the index SNP. Using ANNOVAR software[51], we identified functional variants within each region that were in strong linkage disequilibrium ($r^2 > 0.8$) with the top SNP, including all known coding, splicing, 3′UTR and 5′UTR variants (Supplementary Table 12). To assess for potential functional variation in non-coding regions, we used our recently proposed tissue-specific functional scoring method (FUN-LDA)[11]. Using FUN-LDA, we estimated the posterior probability for each variant in strong LD with the top SNP of being functional across 127 different tissues or cell types profiled by ENCODE and ROADMAP consortia (Supplementary Figs. 6 and 10). To interrogate candidate variants against kidney-specific chromatin landscape, we analyzed regulatory DNase-seq maps paired with RNA-seq gene expression profiles from primary outgrowth cultures of human glomeruli (composed mainly of podocytes and mesangial cells) and renal cortex cultures (composed mainly of tubular cells), as well as chromatin conformation (Hi-C) maps from freshly isolated human glomeruli (Supplementary Figs. 9 and 11)[14]. To test for eQTL effects in kidney tissue, we used gene expression data from the NEPTUNE study[13]. This dataset is comprised of whole genome DNA sequence data and

genome-wide transcriptome data (Affymetrix 2.1 ST chips) performed on micro-dissected glomerular ($N = 136$) and tubulointerstitial ($N = 166$) tissue compartments from kidney biopsies of patients with nephrotic syndrome. For each locus, we tested the index SNP and its high LD SNPs ($r^2 > 0.8$). Testing for cis-eQTL effects involved all transcripts within 1-Mb region centred on each SNP using the additive linear regression with adjustments for age, sex, PEER factors and first 4 PCs of ancestry as described previously[13]. In addition to kidney tissue, all loci were similarly interrogated for eQTL effects in the GTEx database version 8. Because *NFKB1* and *IRF4* both encode transcription factors, we also explored their potential binding in close proximity of each other or *PLA2R1* gene. Based on the Chip-seq data for all five subunits of NFκB complex in immortalized lymphocytes (GSE55105)[29], we found that the top SNPs at *PLA2R1* and *IRF4* loci intersect potential NFκB complex binding site (Supplementary Figs. 8 and 13). In addition, rs230492, a variant in strong LD (r2=0.94) with the top SNP at the *NFKB1* locus intersects a potential IRF4 binding site based on the Chip-seq data of IRF4 (GEO: GSM803390).

**Pleiotropy analysis.** We used the latest GWAS catalogue data to perform systematic cross-annotation of our top risk alleles against all other published GWAS findings. We first identified all genome-wide significant SNPs ($P < 5 \times 10^{-8}$) reported in the catalogue that resided within the genomic regions of association with MN. We assessed the extent of linkage disequilibrium ($r^2$) between these SNPs and the top MN risk alleles based on the combined European and East Asian sequence data from 1000 Genomes (phase 3). We next defined the directionality of pleiotropic effects as either concordant or opposed in relationship to the MN risk alleles. In addition, we queried each qualifying SNP from the catalogue against our genome-wide summary statistics to extract the odds ratios and $P$-values for associations with MN. We defined overlapping susceptibility alleles if $r^2$ exceeded 0.2 (Supplementary Table 13). Lastly, we constructed a susceptibility overlap map that connects each of the MN loci to the previously associated GWAS traits and highlights associations with SNPs in high LD with the top MN signals (Fig. 5a). The map was visualized with Cytoscape v.3.6 software.

**Podocyte gene annotations of GWAS loci.** To search for potential novel podocyte antigens encoded by our suggestive loci, we interrogated each locus against a podocyte-specific gene list predicted with in silico "nanodissection" approach (Supplementary Table 14). This computational approach used Affymetrix gene expression data from micro-dissected glomerular and tubulo-interstitial compartments of 452 renal biopsies[52]. In addition, we cross-annotated our positional candidates against the list of native podocyte proteins discovered by proteomic profiling of mouse podocytes[53]. All suggestive loci were additionally tested for multiplicative interactions with classical alleles, and the top most significant interactions were summarized in Supplementary Table 14. Lastly, we annotated all positional candidate genes using manual PubMed literature searches to prioritize genes with a previously established role in podocyte biology.

**GR analysis.** To estimate the cumulative effect of independently significant GWAS loci, we used a GRS approach. We first identified SNPs with independent contributions to MN risk at each locus using stepwise conditional analyses. Next, we tested for multiplicative interactions among those independently associated SNPs. We then built a logistic regression model including all independent SNPs along with their significant interaction terms to derive mutually adjusted effect sizes (Supplementary Table 15). Because we observed ethnicity-specific signals at the HLA locus, we generated ethnicity-specific for East Asians and Europeans separately. The ethnicity-specific risk scores were defined as a weighted sum of independent risk alleles and their significant interaction terms, weighted by their mutually adjusted effect sizes. The GRS was standardized using a Z-score transformation based on the mean and standard deviation for the distribution of ethnically matched controls, so that the standardized GRS was reflective of the distance between the raw score and the control mean in units of standard deviation. The final formulation of the GRS is as follows:

$$
\begin{aligned}
\text{East Asian GRS}_i = & [0.69173 \times d(rs9269027 - A)_i + 1.23685 \times d(rs1974461 - T)_i + 0.36687 \\
& \times d(rs6707458 - G)_i + 0.25098 \times d(rs230540 - C)_i + 0.39127 \\
& \times d(rs9405192 - G)_i + 0.48798 \times d(rs9269027 - A)_i \times d(rs6707458 - G)_i \\
& - \text{East Asian Control Mean GRS}] / \text{ East Asian Control SD GRS}
\end{aligned} \quad (2)
$$

where $GRS_i$ = genetic risk score for individual $i$, $d_i$ = dosage of risk allele (from 0 to 2) for individual $i$, East Asian Control Mean GRS = 1.6804, East Asian Control SD GRS = 1.003

$$
\begin{aligned}
\text{European GRS}_i = & [0.34945 \times d(rs9271541 - C)_i + 0.67919 \times d(rs9265949 - T)_i \\
& + 0.30707 \times d(rs2858309 - C)_i + 0.34601 \times d(rs6707458 - G)_i \\
& + 0.17450 \times d(rs230540 - C)_i + 0.18343 \times d(rs9405192 - G)_i \\
& + 0.33782 \times d(rs9271541 - C)_i \times d(rs6707458 - G)_i \\
& - \text{European Control Mean GRS}] / \text{European Control SD GRS}
\end{aligned} \quad (3)
$$

where $GRS_i$ = genetic risk score for individual $i$, $d_i$ = dosage of risk allele (from 0 to 2) for individual $i$, European GRS control mean = 1.5089, European GRS control SD = 0.8202.

The percentage of the total variance in disease risk explained was estimated using Nagelkerke's pseudo $R^2$ from the logistic regression model with the standardized GRS as a predictor and case-control status as an outcome. The performances of the ethnicity-specific GRS were estimated by the AUROC. We performed detailed GRS cut-off analyses in the discovery cohorts by selecting cut-off points on the ROC curve that provide specificities in the range from 95 to 100%. For each cut-off point, we calculate sensitivity, specificity, positive likelihood ratio (LR+), and negative likelihood ratio (LR-). All GRS analyses were implemented in R version 3.3.2.

**CRS formulation.** The CRS was formulated as a weighted sum of the GRS and serum anti-PLA2R antibody levels in U/mL. The weight was determined using logistic regression model, with standardized ethnicity-specific GRS (formulated as described above) and natural log-transformed anti-PLA2R antibody levels as two predictors and case-control status as an outcome. This resulted in the following model:

$$Y_i = \beta_0 + \beta_1 GRS_i + \beta_2 \ln(\alpha PLA2R_i + 0.001) \tag{4}$$

where $GRS_i$ indicates the Z-transformed ethnicity-specific GRS for individual $i$, $\beta_0$ indicates the intercept, $\beta_1$ indicates the effect size of the GRS estimated based on discovery cohorts; $\alpha PLA2R_i$ is the serum level of PLA2R antibodies in U/mL; the constant 0.001 is added to enable log transformation of undetectable (zero) levels; $\beta_2$ represents the effect size for anti-PLA2R antibody positivity estimated based on discovery cohorts. The weight for the $\ln(\alpha PLA2R_i + 0.001)$ term was then defined as follows:

$$\text{Weight} = \frac{\beta_2}{\beta_1} \tag{5}$$

Consequently, the weights for antibody levels were calculated for East Asian and European separately, which resulted in the following Crude CRS formulation:

$$\text{Crude CRS}_i = \begin{cases} GRS_i + 0.4829 \times \ln(\alpha PLA2R_i + 0.001), \text{ if } i \in \text{European} \\ GRS_i + 1.7712 \times \ln(\alpha PLA2R_i + 0.001), \text{ if } i \in \text{East Asian} \end{cases} \tag{6}$$

In the final step, the Crude CRS was Z-transformed using the mean and standard deviation for ethnicity-matched healthy controls:

$$\text{CRS}_i = \begin{cases} (\text{Crude CRS}_i - \text{European Control Mean CRS})/(\text{European Control SD}), \text{ if } i \in \text{European} \\ (\text{Crude CRS}_i - \text{E.Asian Control Mean CRS})/(\text{E.Asian Control SD}), \text{ if } i \in \text{East Asian} \end{cases} \tag{7}$$

where European Control Mean CRS = −1.4982, European Control SD = 1.4354, E. Asian Control Mean CRS = 0.3724, E. Asian Control SD = 2.7503.

The CRS performance was estimated by the area under the receiver operating curve (AUROC). Similar to GRS, we explored different CRS cut-offs that maximized specificity in the range 95 to 100%. For each cut-off, we calculated sensitivity, specificity, positive likelihood ratio (LR+), and negative likelihood ratio (LR−). The integrated discrimination improvement (IDI) and net reclassification improvement (NRI)[54] were calculated comparing the CRS test to the serum PLA2R antibody test alone. All CRS analyses were implemented in R version 3.3.2 (CRAN).

**European GWAS validation cohorts.** For the purpose of GRS validation, we utilized three previously published external GWAS cohorts of European ancestry: the UK, French, and Dutch Validation Cohorts[6]. These cohorts were composed of biopsy-documented cases of primary MN and ethnicity-matched healthy controls, totalling 2,887 individuals (550 cases and 2337 controls, see Supplementary Methods for details). For all three cohorts, we obtained primary genotype data after quality control analysis as published previously[6] and performed phasing and imputation using Eagle v.2.3 and Minimac3 with European populations of Phase 3 1000 Genome Project as a reference. The cases and controls were imputed jointly. The GRS for each individual was determined using genotype dosages for the SNPs included in the score. The performance of GRS was then analyzed individually in each cohort, and in all cohorts combined (Supplementary Tables 17–19).

**NEPTUNE GRS validation cohorts.** The Nephrotic Syndrome Study Network (NEPTUNE) is a prospective, longitudinal cohort recruiting participants with nephrotic syndrome at the time of first kidney biopsy[8]; primary disease diagnoses of MN, FSGS, MCD, and IgAN were determined by a central pathology review. All NEPTUNE participants underwent low-depth whole genome sequencing as described in the Supplementary Methods. The GRS was successfully determined for $N = 475$ NEPTUNE participants. This included 89 cases with the diagnosis of primary MN and 386 disease controls, including 184 with FSGS, 164 with MCD, and 38 with IgAN. Our pre-specified primary GRS validation involved 180 NEPTUNE participants of European ancestry (46 cases and 134 disease controls, Supplementary Tables 17–18). In secondary analyses, we extended our validation studies to the entire multiethnic cohort of 475 NEPTUNE participants (Supplementary Tables 20–21), including subgroup analyses of 133 individuals of African American ancestry (18 cases and 115 disease controls) and 94 individuals of Hispanic/Latino American ancestry (18 cases and 76 disease controls). The

numbers of NEPTUNE participants in other ancestral groups were too small for a meaningful analysis.

**PLA2R antibody testing.** In total, we determined serum antibody levels in $N = 2331$ study participants with genetic data ($N = 1488$ cases, $N = 300$ healthy controls, and $N = 543$ disease controls) across all cohorts and ethnicities. The ancestry-matched diseased controls were recruited among patients commonly presenting with nephrotic syndrome, including FSGS, MCD, IgAN. For all MN cases and disease controls, serum samples were obtained near or at the time of kidney biopsy and any samples obtained more than six months after the biopsy were excluded from the analysis. All individuals that underwent antibody testing had matched genetic data, enabling derivation and diagnostic testing of the CRS.

We performed a standardized measurement of serum anti-PLA2R Ab levels using the anti-PLA2R ELISA (IgG) test kit (EUROIMMUN Medizinische Labordiagnostika AG), which employs the indirect ELISA methodology. The kit includes a 96-well microplate pre-coated with PLA2R, 5 calibrators (2, 20, 100, 500, and 1500 U/mL respectively), positive and negative control samples, peroxidase-labelled anti-human IgG (rabbit) enzyme conjugate, kit specific sample and wash buffers, Chromogen/substrate solution (TMB/$H_2O_2$), and stop solution. The assay was run as per the protocol included with the kit. A 5-point calibrated analysis was used to calculate the results for each assay performed. A standard curve was generated based on the spectrophotometric reading of the five calibrators included on each microplate. As recommended by EUROIMMUN, the sample was called positive if antibody level was ≥20 U/mL.

In the discovery stage, we analyzed sera for a total of $N = 459$ East Asian and $N = 1034$ European participants. The European cohorts included 810 cases with MN, 99 healthy controls, and 125 disease controls (37 FSGS and 88 IgAN). The East Asian cohorts included 304 cases with MN, 56 healthy controls, and 99 disease controls (52 FSGS and 47 IgAN). The disease controls were not included in the GWAS discovery analysis, but were added to test for disease specificity of the serologic test. We note that one East Asian disease control with a clinical diagnosis of IgAN tested anti-PLA2R antibody positive at high titer; follow-up pathology review found evidence of previously unrecognized sub-epithelial deposits diagnostic of MN in addition to IgA-dominant mesangial deposits; this individual was subsequently removed from the analysis.

In the validation phase, we analyzed sera for a total of $N = 540$ individuals including European case-control cohorts ($N = 248$ cases and $N = 145$ healthy controls) and a total of 147 European NEPTUNE participants ($N = 36$ cases and $N = 111$ disease controls). In secondary analysis, we extended testing to additional 180 NEPTUNE participants of non-European ancestry ($N = 28$ cases and $N = 152$ controls). This included 103 NEPTUNE participants of African American ancestry (16 cases and 87 disease controls) and 77 participants of Hispanic/Latino ancestry (12 cases and 65 disease controls). These numbers are smaller compared to the GRS validation cohorts, since not all NEPTUNE participants had sera sampled within 6 months of biopsy.

**Testing for phenotypic correlations of the GRS.** Using extensive clinical information available for our discovery cohorts, we investigated correlations between GRS and clinical traits from the time of kidney biopsy including age at diagnosis, 24-h proteinuria (P24), estimated glomerular filtration rate (eGFR), serum albumin (Alb) and serum anti-PLA2R1 antibody level (Supplementary Table 16). The eGFR was estimated based on serum creatinine level using the Modification of Diet in Renal Disease (MDRD) equation[55]. The proteinuria was quantified using spot urine protein-to-creatinine ratios. The values of proteinuria and eGFR were normalized by natural log-transformation. The serum albumin and anti-PLA2R1 antibody levels also required natural log-transformation. For each quantitative trait, we built a linear regression model with GRS as a predictor and each corresponding trait as an outcome. For dichotomous traits (anti-PLA2R seropositivity or presence of nephrotic range proteinuria), we used logistic regression with a GRS as a predictor and each binary trait as an outcome. The association analysis for age at diagnosis (biopsy) was performed before and after adjustment for sex and ancestry. The tests of proteinuria, eGFR, serum albumin and serum anti-PLA2R1 antibody levels were carried out before and after controlling for age, sex and ancestry. Statistical analyses were implemented in R version 3.3.2.

**SNP-based heritability.** We estimated the SNP-based heritability of MN in East Asians and Europeans using the GCTA-GREML algorithm[30,56]. For this analysis, we included three European cohorts (European discovery 1, European discovery 2 and Turkish discovery) and three East Asian cohorts (Chinese, Korean and Japanese discovery) that had primary genotype data available for joint heritability analysis. We first estimated pairwise genetic relationship matrix between all individuals using autosomal SNPs. With the GCTA software[57], we next estimated the disease variance explained by all autosomal SNPs. We transformed the estimate assuming an underlying liability scale and disease prevalence of 0.001. We derived a standard error (SE) and 95% confidence interval for each estimate.

**Reporting summary.** Further information on research design is available in the Nature Research Reporting Summary linked to this article.

## Data availability

All genome-wide summary statistics, including those presented in Fig. 1, are freely available for download on our lab website: www.columbiamedicine.org/divisions/kiryluk/resources.php. The calculations of genetic risk score (GRS) and combined risk score (CRS) are implemented in the form of an online risk calculator, which is also freely available on our lab website. The PAGE consortium control genotype data is available on dbGAP under accession number phs000356.v2.p1. Primary genotype data for the European-1 discovery cohort is available under dbGAP accession number phs001984.v1.p1. Our IRB determined that the use of this dataset is restricted to genetic studies of kidney disease. Because of consent restrictions and/or country-specific privacy laws, we are unable to share primary genotype data on dbGAP for other cohorts. All data and summary statistics are available from the corresponding authors upon reasonable request.

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

## Acknowledgements

We are grateful to all study participants across multiple nephrology centres worldwide for their contributions to this work. This work was supported by the following institutions, grants and funding agencies in the US: Columbia University, Columbia Glomerular Center, National Institute for Diabetes and Digestive Kidney Diseases (NIDDK) grants RC2-DK116690 (K.K., M.K.), R01-DK105124 (K.K.), R01-DK097053 (L.H.B., M.K.), R01-DK108805 (M.G.S.), and National Institute on Minority Health and Health Disparities (NIMHD) grant R01-MD009223 (K.K., A.G.G., A.B.). M.G.S. is additionally supported by the Charles Woodson Clinical Research Fund. The Nephrotic Syndrome Study Network Consortium (NEPTUNE), U54-DK-083912, is a part of the National Institutes of Health (NIH) Rare Disease Clinical Research Network (RDCRN), supported through a collaboration between the Office of Rare Diseases Research, National Center for Advancing Translational Sciences (NCATS) and the National Institute of Diabetes, Digestive, and Kidney Diseases (NIDDK). Additional funding and/or programmatic support for this project has also been provided by the University of Michigan, the NephCure Kidney International and the Halpin Foundation. The recruitment and analysis of the Chinese cohorts were supported by the National Key Research and Development Program of China (2016YFC0904100), Natural Science Foundation of China to the Innovation Research Group (81621092), National Natural Science Foundation of China (No. 81870460, 81570598), Science and Technology Innovation Action Plan of Shanghai Science and Technology Committee (No.17441902200), Shanghai Municipal Education Commission, Gaofeng, Clinical Medicine Grant (No.20152207), Shanghai Jiao Tong University School of Medicine, Multi-Center Clinical Research Project (No.DLY201510), International Cooperation and Exchange Projects of Shanghai Science and Technology Committee (No.14430721000), the Outstanding Young Scholar Award for Zhao Cui (No.81622009), and Shanghai Health and Family Planning Committee Hundred Talents Program for Jingyuan Xie (No.2018BR37). The recruitment of the Korean cohort was supported by the Seoul National University Hospital Human Biobank, a member of the National Biobank of Korea, financed by the Ministry of Health and Welfare, Republic of Korea. P.B. and P.H. acknowledge financial support from MRC project "Autoimmunity in Membranous Nephropathy", grant MR/J010847/1 which funded the sample collection from MN patients across the UK. P.B., P.H. and S.H. acknowledge support from Manchester Academic Health Science Centre (MAHSC 186/200), the Greater Manchester Local Clinical Research Network and Kidneys for Life Charity for supporting research in MN in Manchester. We are grateful to the MENTOR study (clinical trials no. NCT01180036), for contributing blood samples of trial participants. The UK cohort was supported in part by grants from the David and Elaine Potter Charitable Foundation (to S.H.P. and R.K.), St Peter's Trust for Kidney, Bladder and Prostate Research (to D.B., H.C.S., S.H.P. and R.K.), Kids Kidney Research UK and Kidney Research UK (to D.B. and R.K.). The Italian cohorts were supported by the Italian Ministry of Health grant GR-2011-02350438 (G.Z., S.G.) and the Department of Excellence Grant 2018–2022 funded by the Italian Ministry of Education for the Department of Medical Sciences of the University of Turin (A.A.). The recruitment of Polish cases was sponsored by the Polish Kidney Genetics Network (POLYGENES), a collaborative effort between Columbia University and Poznań University of Medical Sciences, Poland. The full list of POLYGENES collaborators can be found in the Supplementary Materials. The GCKD (German Chronic Kidney Disease) study was funded by grants from the German Ministry of Education and Research (BMBF, No. 01ER0804) and the KfH Foundation for Preventive Medicine, with genotyping supported by Bayer Pharma AG. The list of GCKD investigators can be found in the Supplementary Materials. The work of M.W. and A.K. was funded by the CRC 1140 Initiative and by KO 3598/3–1 and CRC 992 (A.K.) of the German Research Foundation. The work of E.H. and R.A.K.S. was funded by the CRC 1192 from the German Research Foundation (Projects B1 and C1). P.R. is a recipient of European Research Council ERC-2012-ADG_20120314 grant 322947, 7th Framework Programme of the European Community contract 2012–305608 (European Consortium for High-Throughput Research in Rare Kidney Diseases), and the National Research Agency grant MNaims (ANR-17- CE17-0012-01). The Dutch studies were supported by grants from the Dutch Kidney Foundation to JMH and JFW (Nierstichting Nederland grant OW08 and grant KJPB11.021). We would like to thank the Population Architecture Using Genomics and Epidemiology (PAGE) consortium, funded by the National Human Genome Research Institute (NHGRI) with co-funding from the NIMHD, for providing population controls for this study. For full acknowledgment of the PAGE consortium, please see Supplementary Materials. The funding sources were not involved in the study design, collection, analysis, and interpretation of data, writing of the report, or in the decision to submit the paper for publication.

## Author contributions

K.K., N.C. and R.K. conceived the study and made the decision to publish the findings; K. K. designed the study and provided overall supervision of the project; K.K. and L. Liu wrote the initial draft of the manuscript; J. Xie coordinated recruitment and genotyping of the Chinese discovery and replication cohorts; N.M. and J. Xie performed quality control, imputation and GWAS association analyses for Asian discovery cohorts; L. Liu performed quality control, imputation and GWAS association analyses for European discovery and validation cohorts, and performed final statistical and bioinformatics analyses, including GWAS meta-analyses, fine-mapping, HLA analyses, functional annotations, risk score analyses, clinical correlation analyses, and generated figures and tables; I.I.-L. consulted on the statistics and functional annotation of GWAS loci. A.G.G. consulted on the study design and critically reviewed the manuscript. J. Xie, X.Y., O.B., L.H.B., H.D., E.H., R.A.K.S., P.B. and R.K. performed serum antibody studies. O.B., Y.L., J.Y.Z., P.K., N.M., R.J.R., M. Bodria, A. Khan, K. Mehl, and F.O. contributed to sample processing, biobanking, DNA extractions, quality control, plating, clinical and genetic data management. S.A. generated kidney compartment-specific maps of chromatin accessibility and interactions. J. Xie, N.C., M-H.Z., Z.C., L.H.R., W.W., Z.L., X.H., X.Y., D.Z., J. Xu, G.L. recruited, genotyped, and clinically characterized the Chinese Discovery and Chinese Replication cohorts. A.P., C.S., M.Z., and F.C. genotyped, imputed, and analyzed GWAS data for the Sardinian Discovery cohort. M.W., K-U.E., and A. Köttgen coordinated the GCKD Study, genotyped, imputed and analyzed GWAS data for the GCKD Discovery cohort. H.S. recruited and clinically characterized the Japanese Discovery cohort. H.L., J.P., B.L.C., Y.S.K., and D.K.K. recruited and clinically characterized the Korean Discovery cohort. Y.C., S.O., A.Y., N.S., H.A., M. Koc, T.B., G.K., S.U.A., and M.S.S. recruited and clinically characterized the Turkish Discovery cohort. M.M., P.A.C., A.S.B., G.B.A., S.S.-C., M. Bodria, F.Z., A.P.-P., M.D., K. Mucha, B.M., B.F., L.P., I.H., E.A., J.B., L.M., B.V., D. Santoro, M. Bonomini, F. Londrino, L.G., J.R., V.T., C.I., S.S., D. Spotti, C.M., P.M., M.G., D.R., S. Granata, G.Z., F. Lugani, G.M.G., I.P., L.A., B. Sprangers, D.C.C., F.C.F., A.A., and F.S. recruited and clinically characterized patients for the European-1 Discovery Cohort. P.H., S.H., S. Gupta, C.C., S.D., N.I., R.J.P., J.C., S.P., D.B., H.C.S., N.A., E.H., R.A.K.S., P.B. and R.K. participated in the recruitment, clinical characterization and generation of genotype data for the European Discovery-2 and the UK Validation cohorts. B. Stengel, H.D., and P.R. contributed the French Validation Cohort. J.M.H., M.J.C., L.A.K., and J.F.M.W. contributed the Dutch Validation Cohort. M.G.S., L.H.M., L.H.B. and M. Kretzler contributed the NEPTUNE validation cohort. R.J.F.L. and E.E.K. contributed the control genotype data for the PAGE cohort. All authors have read and approved the final version of manuscript.

## Competing interests

The authors declare no competing interests.

## Additional information

Jingyuan Xie[1,77]✉, Lili Liu[2,77], Nikol Mladkova[2,77], Yifu Li[2], Hong Ren[1], Weiming Wang[1], Zhao Cui[3,4,5], Li Lin[1], Xiaofan Hu[1], Xialian Yu[1], Jing Xu[1], Gang Liu[3,4,5], Yasar Caliskan[6], Carlo Sidore[7], Olivia Balderes[2], Raphael J. Rosen[2], Monica Bodria[2,8], Francesca Zanoni[2,9], Jun Y. Zhang[2], Priya Krithivasan[2], Karla Mehl[2], Maddalena Marasa[2], Atlas Khan[2], Fatih Ozay[2], Pietro A. Canetta[2], Andrew S. Bomback[2], Gerald B. Appel[2], Simone Sanna-Cherchi[2], Matthew G. Sampson[10], Laura H. Mariani[11,12], Agnieszka Perkowska-Ptasinska[13], Magdalena Durlik[13], Krzysztof Mucha[14,15], Barbara Moszczuk[14], Bartosz Foroncewicz[14], Leszek Pączek[14,15], Ireneusz Habura[16], Elisabet Ars[17], Jose Ballarin[17], Laila-Yasmin Mani[18], Bruno Vogt[18], Savas Ozturk[19], Abdülmecit Yildiz[20], Nurhan Seyahi[21], Hakki Arikan[22], Mehmet Koc[22], Taner Basturk[23], Gonca Karahan[24], Sebahat Usta Akgul[24], Mehmet Sukru Sever[6], Dan Zhang[25], Domenico Santoro[26], Mario Bonomini[27], Francesco Londrino[28], Loreto Gesualdo[29], Jana Reiterova[30], Vladimir Tesar[30], Claudia Izzi[31,32], Silvana Savoldi[33], Donatella Spotti[34], Carmelita Marcantoni[35], Piergiorgio Messa[9], Marco Galliani[36], Dario Roccatello[37], Simona Granata[38], Gianluigi Zaza[38], Francesca Lugani[39], GianMarco Ghiggeri[39], Isabella Pisani[8], Landino Allegri[8], Ben Sprangers[40,41], Jin-Ho Park[42], BeLong Cho[42,43], Yon Su Kim[44,45], Dong Ki Kim[45,46], Hitoshi Suzuki[47], Antonio Amoroso[48], Daniel C. Cattran[49], Fernando C. Fervenza[50], Antonello Pani[51], Patrick Hamilton[52], Shelly Harris[52], Sanjana Gupta[53], Chris Cheshire[53], Stephanie Dufek[53], Naomi Issler[53], Ruth J. Pepper[53], John Connolly[53], Stephen Powis[53], Detlef Bockenhauer[53], Horia C. Stanescu[53], Neil Ashman[54], Ruth J.F. Loos[55,56,57], Eimear E. Kenny[55,58,59], Matthias Wuttke[60], Kai-Uwe Eckardt[61,62], Anna Köttgen[60], Julia M. Hofstra[63], Marieke J.H. Coenen[64], Lambertus A. Kiemeney[65], Shreeram Akilesh[66], Matthias Kretzler[11], Lawrence H. Beck[67], Benedicte Stengel[68,69], Hanna Debiec[70], Pierre Ronco[70,71], Jack F.M. Wetzels[63], Magdalena Zoledziewska[7], Francesco Cucca[7], Iuliana Ionita-Laza[72], Hajeong Lee[45,46], Elion Hoxha[73], Rolf A.K. Stahl[73], Paul Brenchley[74], Francesco Scolari[31,75], Ming-hui Zhao[3,4,5,76], Ali G. Gharavi[2], Robert Kleta[53,77]✉, Nan Chen[1,77] & Krzysztof Kiryluk[2,77]✉

[1]Department of Nephrology, Institute of Nephrology, Shanghai Ruijin Hospital, Shanghai Jiao Tong University School of Medicine, Shanghai, China. [2]Department of Medicine, Division of Nephrology, Columbia University, College of Physicians & Surgeons, New York, USA. [3]Renal Division, Department of Medicine, Peking University First Hospital, Beijing, China. [4]Institute of Nephrology, Peking University, Beijing, China. [5]Key Laboratory of Renal Disease, Ministry of Health of China, and Key Laboratory of CKD Prevention and Treatment, Ministry of Education of China, Beijng, China. [6]Division of Nephrology, Department of Internal Medicine, Istanbul School of Medicine, Istanbul University, Istanbul, Turkey. [7]Istituto di Ricerca Genetica e Biomedica, Consiglio Nazionale delle Ricerche, Monserrato, Cagliari, Italy. [8]Department of Medicine and Surgery, University of Parma, Parma, Italy. [9]Nephrology Dialysis and Kidney Transplant Unit, Fundazione IRCCS Ca' Granda Ospedale Maggiore Policlinico, Università degli studi di Milano, Milan, Italy. [10]Department of Pediatrics-Nephrology, University of Michigan School of Medicine, Ann Arbor, MI, USA. [11]Division of Nephrology, Department of Medicine, University of Michigan, Ann Arbor, MI, USA. [12]Arbor Research Collaborative for Health, Ann Arbor, MI, USA. [13]Department of Transplantology, Nephrology and Internal Diseases, Medical University of Warsaw, Warsaw, Poland. [14]Department of Immunology, Transplantology and Internal Diseases, Medical University of Warsaw, Warsaw, Poland. [15]Institute of Biochemistry and Biophysics, Polish Academy of Sciences, Warsaw, Poland. [16]Department of Nephrology, University Hospital of Karol Marcinkowski in Zielona Góra, Zielona Góra, Poland. [17]Molecular Biology Laboratory and Nephrology Department, Fundació Puigvert, Instituto de Investigaciones Biomédicas Sant Pau, Universitat Autònoma de Barcelona, REDINREN, IISCIII, Barcelona, Spain. [18]Department of Nephrology and Hypertension, Bern University Hospital, University of Bern, Bern, Switzerland. [19]Nephrology Clinic, Haseki Training and Research Hospital, Istanbul, Turkey. [20]Department of Nephrology, Uludag University Faculty of Medicine, Bursa, Turkey. [21]Division of Nephrology, Department of Internal Medicine, Cerrahpasa Medical Faculty, Istanbul University - Cerrahpasa, Istanbul, Turkey. [22]Division of Nephrology, Department of Internal Medicine, Marmara University School of Medicine, Istanbul, Turkey. [23]Department of Nephrology, Sisli Hamidiye Etfal Training and Research Hospital, Istanbul, Turkey. [24]Department of Medical Biology, Istanbul School of Medicine, Istanbul University, Istanbul, Turkey. [25]Department of Nephrology, Xin Hua Hospital Affiliated to Shanghai Jiao Tong University School of Medicine, Shanghai, China. [26]Department of Cinical and Experimental Medicine, Unit of Nephrology, University of Messina, Messina, Italy. [27]Department of Medicine, University of Chieti-Pescara, SS. Annunziata Hospital, Chieti, Italy. [28]S. Andrea Hospital, La Spezia, Italy. [29]University of Bari, Bari, Italy. [30]Department of Nephrology, 1st Faculty of Medicine and General University Hospital, Charles University, Prague, Czech Republic. [31]Second Division of Nephrology, ASST-Spedali Civili di Brescia Presidio di Montichiari, Brescia, Italy. [32]Department of Obstetrics and Gynecology, ASST Spedali Civili di Brescia, Brescia, Italy. [33]Unit of Nephrology and Dialysis ASL TO4, Cirié, Turin, Italy. [34]San Raffaele Hospital, Milan, Italy. [35]Cannizzaro Hospital, Catania, Italy. [36]Sandro Pertini Hospital, Rome, Italy. [37]San Giovanni Bosco Hospital (ERK-net Member) and University of Turin, Turin, Italy. [38]Renal Unit, Department of Medicine,

University of Verona, Verona, Italy. [39]Division of Nephrology, Dialysis, Transplantation, IRCCS Giannina Gaslini, Genoa, Italy. [40]Department of Microbiology and Immunology, Laboratory of Molecular Immunology, Rega Institute, KU, Leuven, Belgium. [41]Department of Nephrology, University Hospitals Leuven, Leuven, Belgium. [42]Department of Family Medicine, Seoul National University College of Medicine and Seoul National University Hospital, Seoul, Korea. [43]Institute on Aging, Seoul National University College of Medicine, Seoul, Korea. [44]Biomedical Sciences, Seoul National University College of Medicine, Seoul, Korea. [45]Kidney Research Institute, Seoul National University College of Medicine, Seoul, Korea. [46]Internal Medicine, Seoul National University College of Medicine, Seoul, Korea. [47]Department of Nephrology, Juntendo University Faculty of Medicine, Tokyo, Japan. [48]Department of Medical Sciences, University of Torino and Immunogenetics and Transplant Biology Service, University Hospital "Città della Salute e della Scienza di Torino", Turin, Italy. [49]Department of Nephrology, University of Toronto, Toronto General Hospital, Toronto, ON, Canada. [50]Division of Nephrology and Hypertension, Mayo Clinic, Rochester, MN, USA. [51]Department of Nephrology and Dialysis, G. Brotzu Hospital, Cagliari, Italy. [52]Manchester Institute of Nephrology and Transplantation, Manchester University Hospitals NHS Trust, Manchester, UK. [53]Department of Nephrology, Division of Medicine, University College London, London, UK. [54]Renal Unit, Royal London Hospital, Barts Health, Whitechapel, London, UK. [55]The Charles Bronfman Institute for Personalized Medicine, Icahn School of Medicine at Mount Sinai, New York, NY, USA. [56]The Genetics of Obesity and Related Metabolic Traits Program, The Icahn School of Medicine at Mount Sinai, New York, NY, USA. [57]The Mindich Child Health and Development Institute, Icahn School of Medicine at Mount Sinai, New York, NY, USA. [58]Department of Genetics and Genomic Sciences, Mount Sinai Health System, New York, NY, USA. [59]Center for Population Genomic Health, Icahn School of Medicine at Mount Sinai, New York, NY, USA. [60]Institute of Genetic Epidemiology, Dep. of Biometry, Epidemiology, and Medical Bioinformatics, Faculty of Medicine and Medical Center – University of Freiburg, Freiburg, Germany. [61]Department of Nephrology and Medical Intensive Care, Charité – Universitätsmedizin Berlin, Berlin, Germany. [62]Department of Nephrology and Hypertension, Friedrich-Alexander-Universität, Erlangen, Germany. [63]Department of Nephrology, Radboud University Medical Center, Nijmegen, The Netherlands. [64]Department of Human Genetics, Radboud University Medical Centre, Radboud Institute for Health Sciences, Nijmegen, The Netherlands. [65]Department of Epidemiology, Biostatistics & HTA, Radboud University Medical Center, Nijmegen, The Netherlands. [66]Department of Anatomic Pathology, University of Washington, Seattle, USA. [67]Department of Medicine, Renal Section, Boston University School of Medicine and Boston Medical Center, Boston, MA, USA. [68]Institut National de la Santé et de la Recherche Médicale, Centre for Research in Epidemiology and Population Health, Villejuif, France. [69]University Paris-Sud, Villejuif, France. [70]Sorbonne Université, Pierre and Marie Curie University Paris 06, Paris, France. [71]Institut National de la Santé et de la Recherche Médicale, Unité Mixte de Recherche (UMR) 1155, Paris, France. [72]Department of Biostatistics, Mailman School of Public Health, Columbia University, New York, NY, USA. [73]III Department of Medicine, University Medical Center Hamburg-Eppendorf, Hamburg, Germany. [74]Faculty of Biology, Medicine, Health, University of Manchester, Manchester, UK. [75]University of Brescia, Brescia, Italy. [76]Peking-Tsinghua Center for Life Sciences, Beijing, China. [77]These authors contributed equally: Jingyuan Xie, Lili Liu, Nikol Mladkova, Robert Kleta, Nan Chen, Krzysztof Kiryluk. ✉email: nephroxie@163.com; r.kleta@ucl.ac.uk; kk473@columbia.edu

