## [Peer Review File · Nature Communications]

Reviewers' Comments:

Reviewer #1:

Remarks to the Author:

Membranous Nephropathy (MN) is a relatively rare nephropathy; usually presenting with proteinuria/nephrotic syndrome in middle-aged patients. There is paucity of data on the genetic architecture of this disease and the authors should be congratulated on assembling a relatively large number of patients with biopsy-proven membranous nephropathy for the purpose of GWAS.

1) The refining of the association signal within PLA2R1 locus is interesting and important with rs17241973 showing the directionally opposite allelic effect on the expression of PLA2R1 between the kidney and non-renal tissue. I would have liked to see a little bit more contemplation of this finding in discussion [in the context of other types of kidney-specific GWAS eQTL signals (ie. Those with the difference in eGene identity between the kidney and non-kidney tissues – see ie. PMID: 30467309).]

2) I understand that rs17831251 showed no effect on the renal expression of PLA2R1 in NEPTUNE study?

3) Is NFKB1 a renal eGene for rs230540? What were the results of cis-eQTL analysis in NEPTUNE?

4) Do the new variants for MN uncovered by this GWAS map onto specific cell-types in single cell maps of the human kidney?

5) Is there an overlap between any of the newly uncovered loci and those identified in GWAS of eGFR/kidney disease (see i.e. PMID: 31152163)?

Reviewer #2:

Remarks to the Author:

Xie, Liu and Mladkova et al present a large GWAS of membranous nephropathy. It combines eight cohorts from European and East Asian ancestries and revealed two novel risk loci that are associated with MN. The authors made an in-depth effort to dissect ancestry-specific genetic signals related to MN, and advocated genetic risk score can be incorporated to assist future non-invasive diagnostic of MN and offer opportunities for drug repurposing from IBD.

Overall, I think the presented study makes an important addition to the pathogenesis of MN. However, I would like to see further analyses to strengthen the claims made in the paper.

Major comments:

1. I have major concerns on the HLA analysis particularly due to different sizes and qualities of the imputation reference panels (5,225 versus 268). These differences will impact the downstream analyses and claims about heterogenous signals observed in two ethnic groups.

(a) The authors claim that the European signal was boarder, spanning over the MHC region, but it could be a result from fewer variants passed imputation threshold in East Asians. The authors should state clearly what is the number of variants being analyzed in each cohort before and after imputation.

(b) Similarly, for the classical HLA alleles and amino acid analyses, observed results could be due to differences in the imputation panels. In particular, that fact that DRB1*15:01 has no significant and opposite risk effect in Europeans is confusing. Do the authors suggest that the same HLA allele is risk in East Asian population and protective in Europeans?

I suggest the authors to look for a bigger imputation reference panel for East Asians (e.g. Zhou et. al Nature Genetics 2016), and re-examine their results. At minimum, they can use the same imputation reference panel (T1DGC, which has OK imputation quality for East Asians) for both ancestries and test whether the observed differences are due to differences in the reference panel or true biological difference between two ethnicities.

2. The authors made a claim that a single risk haplotype per locus is shared between two

ancestries. Why don't they perform a fine-mapping analysis to find the credible set for each region, and test if the causal variants are likely to be shared among two ethnic groups.

3. The authors did a significant amount of work to investigate the genetic differences between the two ethnic groups in MN. Can they also perform a genome-wide genetic correlation test, such as presented in Brown et al. AJHG, 2016, to formally test the trans-ethnic correlation.

4. PC analysis.

(a) PCs presented in SF1 looks atypical for homogenous populations. Specifically: SF1 (A) Chinese discovery cohort shows stratification which looks like a result of not removing high LD regions in the genome. Can the authors explain more in detail how PC analysis was performed across these cohorts? Are they uniformly done across all eight cohorts?

(b) I would like to see all GWAS samples projected onto the same global PC loadings (e.g., those from 1000 Genomes). This will add another layer of evidence that all individuals are indeed of Asian and European ancestry. For example, I am less sure that the Turkish cohort can be grouped into 'European cohorts' without a global PCA.

5. cis-eQTL effect of rs17241973 and PLA2R1 are over-stated. A p-value of 0.024 will not hold after correcting for multiple testing. The authors can either perform a permutation or use $0.05/(\text{number of independent snps or number of total snps in the cis-region})$ to obtain an appropriate significant threshold. The statistics of rs17241973 eQTL in other tissues are also not presented. I am therefore not convinced there is a kidney-specific effect without further supporting evidence.

6. Were covariates such as age and gender also included in the association study? Can the authors provide more details of how they define 'significant' PCs to include in their logistic regression?

7. Total genetic heritability of MN was estimated using LDSC, which excludes the HLA region. However, in the GRS analyses, the authors included the HLA variants. Therefore the reported number of how much total heritability that GRS explained is incorrect.

8. To show what genetic risk can add to diagnostic performance, I would like to see what is the predictive power of using serum anti-PLA2R antibody levels alone in Figure 5 and 6. I am also interested in seeing what is the impact on the predictive power when using a transethnic GRS in the validation cohorts?

9. The authors claim that there is a high concordance rate between MN and IBD. I would like to see a formal analysis using LDSC (Bulik-Sullivan and Finucane, et al. Nature Genetics, 2015) to provide an estimate for the genetic correlation between the two traits.

Minor comments:

1. There is a disagreement between genotyping platform described for the Chinese discovery cohort between Table 1 (MEGA chip) and in Supplementary Material (Zhouhua-8).

2. I would like to see intensity cluster plots of the genotyped variants that have the strongest association in the claimed risk loci. Particularly for the two novel risk loci (NFKB1 and IRF4).

3. No reference for using the GTEx data.

4. In the sentence "rs17241973 exhibits a strong cis-eQTL effect on PLA2R1 expressions", no detailed statistics were provided.

5. No standard errors are presented around h^2_g estimates. And since the authors have access to the raw genotype data, why don't they apply methods such as GCTA to obtain h^2_g with a smaller

standard error?

6. Can authors apply an Asian-specific or trans-ethnic genetic risk score, and test its performance in Hispanic Americans versus African Americans in the NEPTUNE study?

Reviewer #3:

Remarks to the Author:

The paper reports results from a large GWAS of membranous nephropathy (MN), a glomerular disease with autoimmune causes, in individuals of European and East Asian ancestry. The main findings are the identification of two novel loci at NFKB1 and IRF4, and fine-map of the PLA2R1 locus. The authors performed imputation specific for the HLA region and identified several ancestry-specific independent variants associated with MN. In addition, they explored gene-gene interactions among known loci. A clever component of the study is the clinical and diagnosis implications of genotypes including anti- PLA2R1 available in a subset of patients. The paper is well written and has important information for this uncommon disease, in addition of being the larger sample so far collected on this disease. There are several aspects that need clarification, including more details on the source of controls (normal individuals, individuals with kidney disease of other etiologies?). If there is variation on the source of controls, sensitivity analyses will be needed. In addition, the replication/validation studies are not well-described and seem to include studies of patients with a variety of kidney disease instead of only MN. Additional comments/suggestions are listed below:

Title is not accurate and should be modified.

Are there differences in MN prevalence across ancestries to justify using just samples of European and East Asian ancestry? If so, please include this in the introduction. Also explain why other ancestries were not included.

Page 5: under "Study design", sentence "Eight cohorts were genotyped with high density SNP arrays, imputed using the latest whole genome sequence reference panels, and meta-analyzed genome-wide". Be specific on the arrays and reference panels. I think this information should be added to the main text and more specific details included in supplementary material.

Page 6, 2nd paragraph related to complexity of association at the HLA region. Describe differences in the LD structure across ancestries. Some of the results suggest this.

Same page, what are "two- and four-digit classical HLA alleles"?

Page 8, under PLA2R1 locus and its genetic interactions. Please include the N cases/controls by ancestry after the OR. Some of this information is in Table 1 but it is unclear if the study used all the data for analyses (or if there were exclusions based on covariates). I also notice that East Asian ancestry effect estimates had wide confidence intervals suggesting a less precise estimate. How the gene-gene interaction thresholds were estimated? Provide this information in Methods. Which HLA variants were used given differences among ancestry and more than one HLA independent signal at the HLA region. How many tests were performed for the interaction analyses? Please clarify this.

For the sentence " with the risk homozygosity at both loci associated with 89-fold increased odds of disease in East Asians (OR=88.8, 95%CI:38.0-207.3, Pint=7.8x10⁻³) and 14-fold in Europeans (OR=14.1, 95%CI:10.0-22.1, Pint=6.4x10⁻⁵)". Please add the reference genotypes for these results, i.e., compared to xxx.

Page 9, "Our tissue-specific functional scoring method for non-coding variants⁹ prioritized another variant in intron 1, rs17241973 (r²=0.93 with rs17831251) that intersects a putative enhancer element across multiple tissues ", please include information specific to kidney tissue, if the disease affects only kidneys.

Were secondary causes of MN excluded from cases?

Page 10, paragraph on suggestive GWAS loci does not contribute much to the paper and could be removed or moved to the supplementary material.

Page 11, which SNPs were included in the GRS? Please include them in the text. Were all the SNPs present in all studies?

Page 11, under clinical correlations. Were there any differences by MN disease outcome such as remission/response to therapy?

Page 12, for the GRS validation in the European-American NEPTUNE GWAS of incident nephrotic syndrome, please clarify the proportion of MN patients in this study and the reasons for testing a GRS derived from MN in a study of nephrotic syndrome that has other etiologies.

Page 13, Neptune GWAS non-European individuals, same as above. What proportion of individuals had MN and how controls were chosen?

Page 14, 3rd paragraph. The concordant effect of risk loci for MN and IBD is introduced for first time in the discussion. You may want to show some results in the Results section. Also, are these diseases related clinically? This paragraph seems out of place.

Page 15, the comparison of genotypes with ELISA test: is the ELISA test standardized across studies? (same test used in all individuals?). How many individuals were used in these analyses? Table 1, were the controls individuals with CKD without MN or individuals without CKD? Add some information to the footnote of the table.

We would like to thank the expert reviewers for their insightful comments and suggestions. The revised manuscript incorporates these suggestions and we feel is significantly strengthened by the provided feedback. All changes in the text of the manuscript are highlighted in yellow. Our point-by-point responses are provided below:

Reviewer #1:

Membranous Nephropathy (MN) is a relatively rare nephropathy; usually presenting with proteinuria/nephrotic syndrome in middle-aged patients. There is paucity of data on the genetic architecture of this disease and the authors should be congratulated on assembling a relatively large number of patients with biopsy-proven membranous nephropathy for the purpose of GWAS.

We thank the reviewer for recognizing the significance of this large-scale international effort.

1) The refining of the association signal within PLA2R1 locus is interesting and important with rs17241973 showing the directionally opposite allelic effect on the expression of PLA2R1 between the kidney and non-renal tissue. I would have liked to see a little bit more contemplation of this finding in discussion [in the context of other types of kidney-specific GWAS eQTL signals (ie. Those with the difference in eGene identity between the kidney and non-kidney tissues – see ie. PMID: 30467309).]

We thank the reviewer for this suggestion. We expanded the discussion of these effects, including the following sentence: “This finding is consistent with recent analyses of loci for CKD-related traits, demonstrating that among loci that are transcriptionally active in renal tissue, 15.8% of effects are kidney-specific”. Please note that we have also added the newly released GTEx V8 results that now include kidney cortex data and more clearly demonstrate the opposed effects between the kidney and all other GTEx tissues for the top SNPs at the PLA2R1 locus.

2) I understand that rs17831251 showed no effect on the renal expression of PLA2R1 in NEPTUNE study?

The effect of rs17831251 on PLA2R1 in NEPTUNE is depicted in Figure S7 and we now have also added the effects detected in GTEx version 8 (Figure S6). Reassuringly, the effects are direction-consistent for rs17831251 as well as rs17241973 between both NETUNE and GTEx datasets. We have modified the text accordingly.

3) Is NFKB1 a renal eGene for rs230540? What were the results of cis-eQTL analysis in NEPTUNE?

NFKB1 is not a significant renal eGene for rs230540 in NEPTUNE (p=0.59), nor in the latest GTEx release that now includes kidney cortex (p=0.30). This gene is only weakly expressed in various kidney cells including podocytes (see below), but there is a trend for rs230540 risk allele in both GTEx and NEPTUNE that is direction-consistent with that observed for whole blood and T-cells. We note that the sample size of the kidney eQTL datasets is small, limiting the power to detect weaker eQTLs or those specific to a single cell type.

4) Do the new variants for MN uncovered by this GWAS map onto specific cell-types in single cell maps of the human kidney?

We investigated gene expression levels of the three candidate genes encoded by the non-HLA loci in the existing human snRNA-seq dataset², newly added as Figure S11 in the manuscript (also see below). We note that PLA2R1 is strongly expressed (red) only in podocytes, NFKB1 is weakly expressed across all different renal cell types (yellow), while IRF4 does not appear to be expressed in any of the renal cell clusters (blue). Because MN is an autoimmune disease, it is also possible that the NFKB1 and IRF4 risk loci have extra-renal effects in immune cells. Our interrogation of mRNA expression of NFKB1 and IRF4 across all GTEx tissues demonstrates that both NFKB1 and IRF4 are most strongly expressed in EBV-transformed lymphocytes.

5) Is there an overlap between any of the newly uncovered loci and those identified in GWAS of eGFR/kidney disease (see i.e. PMID: 31152163)?

The *NFKB1* locus overlaps with the genome-wide significant locus previously discovered for eGFR that is also significant ($P=1.21E-25$) in the study by Wuttke et al.; the top SNP is in LD with rs230540 and the direction of effect is such that the MN risk allele is associated with lower eGFR (see the pleiotropy map in Figure 4A). None of the other loci were overlapping with GWAS for CKD-related traits. We have now added the reference to the latest CKDGen GWAS to the sentence describing these results: “Notably, the MN risk allele has previously been associated with lower estimated glomerular filtration rate in GWAS of renal function^{3,4}, thus this locus may be more broadly associated with the risk of kidney disease.”

Reviewer #2:

Xie, Liu and Mladkova et al present a large GWAS of membranous nephropathy. It combines eight cohorts from European and East Asian ancestries and revealed two novel risk loci that are associated with MN. The authors made an in-depth effort to dissect ancestry-specific genetic signals related to MN, and advocated genetic risk score can be incorporated to assist future non-invasive diagnostic of MN and offer opportunities for drug repurposing from IBD. Overall, I think the presented study makes an important addition to the pathogenesis of MN. However, I would like to see further analyses to strengthen the claims made in the paper.

We thank the reviewer for his/her appreciation of this work.

Major:

1. I have major concerns on the HLA analysis particularly due to different sizes and qualities of the imputation reference panels (5,225 versus 268). These differences will impact the downstream analyses and claims about heterogenous signals observed in two ethnic groups. (a) The authors claim that the European signal was boarder, spanning over the MHC region, but it could be a result from fewer variants passed imputation threshold in East Asians. The authors should state clearly what is the number of variants being analyzed in each cohort before and after imputation.

Thank you for raising this point. Indeed, both the pre-imputation coverage of the HLA region and post-imputation marker density is lower in East Asian cohorts compared to Europeans (see below). In the association analysis of the HLA region in East Asian cohorts we used ~5,000 common bi-allelic markers imputed at high confidence (MAF>1%, imputation $R^2>0.8$) as compared to ~8,000 markers in Europeans. Accordingly, we have now removed the sentence that compares the span of the association signal between the two ethnicities. Despite the differences in SNP coverage, the numbers of imputed classical HLA alleles per gene are comparable between the cohorts.

Characteristic	Japanese	Korean	Chinese	Turkish	European-1	European-2
Sample Size	439	872	1465	590	2172	2139
Number of SNPs pre-imputation	3955	4224	4280	5535	5616	5561
Number of imputed classical HLA alleles						
HLA-A	12	18	22	28	28	26
HLA-B	23	28	39	43	42	35
HLA-C	22	22	22	32	30	25
HLA-DPA1	5	5	7	6	5	5
HLA-DPB1	14	18	18	19	19	19
HLA-DQA1	12	14	12	12	12	12
HLA-DQB1	15	19	16	18	17	17
HLA-DRB1	23	29	25	27	24	27
Individual AA imputed	370	425	448	353	347	337
Total number of markers imputed	5076	5104	5184	8130	8166	8043

(b) Similarly, for the classical HLA alleles and amino acid analyses, observed results could be due to differences in the imputation panels. In particular, that fact that DRB1*15:01 has no significant and opposite risk effect in Europeans is confusing. Do the authors suggest that the same HLA allele is risk in East Asian population and protective in Europeans? I suggest the authors to look for a bigger imputation reference panel for East Asians (e.g. Zhou et. al Nature Genetics 2016), and re-examine their results. At minimum, they can use the same imputation reference panel (T1DGC, which has OK imputation quality for East Asians) for both ancestries and test whether the observed differences are due to differences in the reference panel or true biological difference between two ethnicities.

Thank you for this suggestion. We agree that the observed ethnicity-specific effects of the HLA-DRB1*1501 allele are unusual. However, our results are fully consistent with two prior targeted studies of the HLA region in Han Chinese (the findings discussed by our group in a recent editorials). Additionally, the panel used in our study has been successfully used in multiple studies and adequately captures common HLA alleles, including DRB1*1501. Nevertheless, we agree that validation of this effect using the largest reference panel available would be reassuring. At the suggestion of the reviewer, we downloaded the latest HLA sequencing data from Zhou et al. including 10,689 healthy individuals of Han Chinese ancestry. We phased the sequence data to create a new reference panel and re-imputed our East Asian cohorts. We reproducibly identified the leading association of DRB1*1501 with increased risk of MN (OR=3.49, P= 3.9E-40). The effect estimate and P-value were comparable with our original analysis (OR=3.81, P=2.0E-49). Similarly, we validated the association of DRB1*0301, the second independent risk allele in East Asians (OR=4.08, P= 6.3E-24 versus the original OR=3.50, P=9.2E-23). Given these results, we are confident that these Asian-specific associations do not represent artifacts of our imputation panels.

2. The authors made a claim that a single risk haplotype per locus is shared between two ancestries. Why don't they perform a fine-mapping analysis to find the credible set for each region, and test if the causal variants are likely to be shared among two ethnicities.

We thank the reviewer for this suggestion – we have now performed fine-mapping of the non-HLA loci using CAVIAR and derived 99% credible sets for each locus based on trans-ethnic as well as ethnicity-specific analyses (see Supplemental Figure S4). The set of predicted causal variants at each locus derived from the trans-ethnic summary statistics overlapped with each of the ethnicity-specific credible sets, providing additional support to the claim that the causal variants at these three loci are likely shared between the two ethnicities. We added the corresponding methods and results to the manuscript.

3. The authors did a significant amount of work to investigate the genetic differences between the two ethnic groups in MN. Can they also perform a genome-wide genetic correlation test, such as presented in Brown et al. AJHG, 2016, to formally test the trans-ethnic correlation.

We estimated the genome-wide genetic correlation between the two ancestries at $r_g=0.24$, but this estimate was not statistically significant (P-value=0.37, se=0.26). We note that the methods for testing genome-wide genetic correlations were designed for highly polygenic traits. Because the genetic architecture of MN does not seem to fit this model, we choose not to include these analyses in the manuscript.

4. PC analysis: (a) PCs presented in SF1 looks atypical for homogenous populations. Specifically: SF1 (A) Chinese discovery cohort shows stratification which looks like a result of not removing high LD regions in the genome. Can the authors explain more in detail how PC analysis was performed across these cohorts? Are they uniformly done across all eight cohorts?

The ancestry for each cohort was analyzed individually using common, high quality, and independent genotyped markers (MAF>1%, genotype call rate>95%, and after pairwise LD pruning). High LD regions including HLA were excluded from these analyses. Because the ancestry analysis used cohort-specific data for marker filtering, the final set of markers used for PCA differed by cohort. We agree that our PCA does pick up a subtle stratification within the Chinese cohort, but this likely reflects ancestral differences between South and North Chinese, two subpopulations

historically separated by the Yangtze river. These differences have previously been described in detail (see Chen et al *AJHG* 2009)⁶, and we have observed a similar substructure in our prior studies based on Han Chinese from Beijing (see Figure S2 in Gharavi et al. *Nat Gen* 2011)⁷. We note that this specific cohort was recruited in the metropolitan region of Shanghai, located at the estuary of the Yangtze River, and thus it is expected to represent a mixture of South and North Chinese. Importantly for our study, the distribution of cases and controls is well balanced across these clusters and we achieve adequate control of genomic inflation with PCA adjustment, indicating that these effects do not confound our association results.

(b) I would like to see all GWAS samples projected onto the same global PC loadings (e.g., those from 1000 Genomes). This will add another layer of evidence that all individuals are indeed of Asian and European ancestry. For example, I am less sure that the Turkish cohort can be grouped into ‘European cohorts’ without a global PCA.

As requested, we now include a PCA plot with 1000G Phase III populations for reference. All Phase 3 reference populations are shaded in gray, and our case-control cohorts are depicted in colors. We observe correct mapping of our cohorts to the European and East Asian population clusters. The Turkish cohort (light blue) is partially overlapping with other European cohorts, and is clearly more proximal to European compared to East Asian or African populations. The Japanese, Chinese, and South Korean case-control cohorts co-cluster closely with East Asian populations of the 1000G, and appear least structured based on this global analysis.

5. cis-eQTL effect of rs17241973 and PLA2R1 are over-stated. A p-value of 0.024 will not hold after correcting for multiple testing. The authors can either perform a permutation or use 0.05/(number of independent snps or number of total snps in the cis-region) to obtain an appropriate significant threshold. The statistics of rs17241973 eQTL in other tissues are also not presented. I am therefore not convinced there is a kidney-specific effect without further supporting evidence.

We now provide additional supporting evidence in the form of multi-tissue eQTL statistics based on the newly released GTEx V8 dataset. Based on these results, it is clear that both rs17241973 and rs17831251 have a strong eQTL effect on PLA2R mRNA levels across multiple tissues (e.g. esophageal mucosa rs17241973: NES=0.446, P=3.9x10⁻⁴²). In fact, 30 out of 49 tissues studied had a direction-consistent eQTL effect at a nominal P<0.05, and only the brain tissues, where PLA2R1 is expressed at a very low level, did not support this effect (see Supplemental Figure S6). Moreover, the latest release of GTEx now includes eQTLs based on kidney cortex (N=73 samples), and this analysis is consistent with our findings from NEPTUNE. Notably, kidney cortex is the only one of 49 GTEx tissues with the opposed eQTL effect, providing further support for a kidney-specific effect of this locus. At the same time, we recognize that the kidney eQTL datasets are considerably smaller compared to other tissues and thus the power to detect significant eQTLs is limited. We have therefore changed the way we describe these findings, and choose to refer to them as “suggestive” throughout the manuscript. In the discussion of limitations, we also added “Moreover, larger glomerular compartment-specific datasets will be needed to confirm the observed eQTL effects”.

6. Were covariates such as age and gender also included in the association study? Can the authors provide more details of how they define ‘significant’ PCs to include in their logistic regression?

Age and gender were not included in the GWAS analysis, but we do not expect confounding of genetic associations by these factors. MN is a rare disease with estimated incidence 1 in 100,000 in the general population⁸. Even in the unlikely

event of a rare control misclassification due to age mismatch (i.e. a rare case classified as a control before any disease manifestation), this would only bias our analysis toward the null hypothesis. Moreover, given the rarity of this phenotype, we would expect this type of misclassification to have a negligible effect on our association results. The significance of ancestry PCs was determined by a Tracy-Widom test as previously proposed (Patterson et al. *PLoS Gen* 2006)⁹ and verified by visual examination of the scree plots. We now include this information in the Supplemental Methods.

7. Total genetic heritability of MN was estimated using LDSC, which excludes the HLA region. However, in the GRS analyses, the authors included the HLA variants. Therefore the reported number of how much total heritability that GRS explained is incorrect.

We agree, the LDSC-based estimates of heritability are not directly comparable to our estimates of the fraction of variance explained by the GRS. Moreover, the LDSC method is designed to estimate heritability under the assumption of additive polygenic effects, and the genetic architecture of MN is clearly violating this assumption. Therefore, we choose to remove the LDSC-based analysis in the present version of the manuscript. When interpreting the effects of GRS, we only refer to the total fraction of disease risk explained by the GRS rather than the fraction of overall SNP-based heritability explained.

8. To show what genetic risk can add to diagnostic performance, I would like to see what is the predictive power of using serum anti-PLA2R antibody levels alone in Figure 5 and 6. I am also interested in seeing what is the impact on the predictive power when using a transethnic GRS in the validation cohorts?

We have now generated additional figures for East Asians and Europeans visualizing ROC AUC for PLA2R antibody levels in comparison to GRS and CRS (new Supplemental Figure S13). We also provide a comparison of the European and rans-ethnic GRS in the European validation cohorts combined (new Supplemental Figure S14).

9. The authors claim that there is a high concordance rate between MN and IBD. I would like to see a formal analysis using LDSC (Bulik-Sullivan and Finucane, et al. *Nature Genetics*, 2015) to provide an estimate for the genetic correlation between the two traits.

We apologize for the confusion. All four MN risk loci described in this study (*PLA2R1*, *IRF4*, *NFKB1*, and *HLA*) are shared between MN and IBD with concordant effects, suggesting potential involvement of common pathogenic pathways in these disorders. By concordance of effects we mean the same direction of allelic effects between MN and IBD across all four lead SNPs at a genome-wide significance. We believe that this statement is correct and based on the analysis presented in Figure 4A and Table S13. The genetic correlation between MN and IBD by LDSC is not significant ($r_g=0.0028$, $P=0.99$), but the LDSC method is not well suited for this analysis given the observed genetic architecture of MN, and we only refer to the sharing of effects at individual genome-wide significant risk loci. We have now re-phrased this sentence to avoid ambiguity.

Minor:

1. There is a disagreement between genotyping platform described for the Chinese discovery cohort between Table 1 (MEGA chip) and in Supplementary Material (Zhouhua-8).

Thank you for pointing this out; we used Zhouhua-8 for the Shanghai cohort and we have corrected Table 1.

2. I would like to see intensity cluster plots of the genotyped variants that have the strongest association in the claimed risk loci. Particularly for the two novel risk loci (*NFKB1* and *IRF4*).

The cluster plots for the top genotyped markers at the *NFKB1* and *IRF4* loci are included below for the two main chips used in the study. We also examined other genotyped SNPs supporting these associations and have not detected any problems with clustering or genotype QC. Notably, both loci were replicated using a completely different targeted genotyping technology (KASP, LGC Genomics) providing reassurance that the observed association signals are correctly called and analyzed.

NFKB1 locus: the lead genotyped SNP (rs1020759, OR=1.25, P=2.7E-10)

IRF4 locus: lead genotyped SNP (rs9405192, OR=1.29, P=1.4E-14)

3. No reference for using the GTEx data.

Thanks for noticing this, the reference to GTEx has been added.

4. In the sentence “rs17241973 exhibits a strong cis-eQTL effect on PLA2R1 expressions”, no detailed statistics were provided.

We now provide GTEx V8 results including all relevant statistics for multiple tissues in the Supplement (see Figure S6).

5. No standard errors are presented around h2g estimates. And since the authors have access to the raw genotype data, why don't they apply methods such as GCTA to obtain h2g with a smaller standard error?

We agree, and we greatly appreciate this suggestion. The GCTA-GREML estimates were more conservative compared to LDSC, and the standard errors were considerably smaller. We have therefore now replaced all LDSC-based analyses with the GCTA-GREML method.

6. Can authors apply an Asian-specific or trans-ethnic genetic risk score, and test its performance in Hispanic Americans versus African Americans in the NEPTUNE study?

We have now provided these results in Supplemental Table S21. European GRS appears to have comparable or superior performance to the trans-ethnic GRS in NEPTUNE African-American and Hispanic/Latino participants. The Asian GRS does not perform well in these ancestral groups.

Reviewer #3:

The paper reports results from a large GWAS of membranous nephropathy (MN), a glomerular disease with autoimmune causes, in individuals of European and East Asian ancestry. The main findings are the identification of two novel loci at NFKB1 and IRF4, and fine-map of the PLA2R1 locus. The authors performed imputation specific for the HLA region and identified several ancestry-specific independent variants associated with MN. In addition, they explored gene-gene interactions among known loci. A clever component of the study is the clinical and diagnosis implications of genotypes including anti- PLA2R1 available in a subset of patients. The paper is well written and has important information for this uncommon disease, in addition of being the larger sample so far collected on this disease. There are several aspects that need clarification, including more details on the source of controls (normal individuals, individuals with kidney disease of other etiologies?). If there is variation on the source of controls, sensitivity analyses will be needed.

We thank the reviewer for these kind comments. We also apologize for not clearly describing the design complexities. All cases in this study were strictly required to have a kidney biopsy diagnosis of MN without any known secondary cause. Except for the GCKD study, all controls used for GWAS discovery were healthy population controls (any individuals with known diagnosis of kidney disease were excluded). Please note that the diseased controls included in Figure 5 (panels G and H) were added only for the purpose of testing the specificity of serologic and genetic tests. These controls were not included in our GWAS discovery analysis.

The GCKD cohort was the only discovery cohort that included CKD controls. This cohort was drawn from a prospective observational study of patients with CKD defined by either an estimated glomerular filtration rate (eGFR) between 30 and 60 mL/min/1.73 m² or overt proteinuria upon study inclusion. We identified idiopathic MN as the leading cause of CKD in 151 of 5217 GCKD patients, 147 of whom had a kidney biopsy confirmation of the diagnosis. We selected 1,655 ancestry-matched GCKD patients for whom CKD etiology was clearly assigned to a non-MN cause, including nephrosclerosis, infections, tumor nephrectomies, interstitial nephritis and vascular diseases, as previously described¹⁰. We now include this information in the revised Study Design section and the Supplemental Methods. We note that with the sample size of 147 MN cases vs. 1,655 non-MN CKD controls, the overall contribution of the GCKD cohort to our study is relatively small; when GCKD cohort is excluded from the meta-analysis, all GWAS loci remained genome-wide significant.

In addition, the replication/validation studies are not well-described and seem to include studies of patients with a variety of kidney disease instead of only MN.

We apologize again for not describing these complex validation studies more clearly. The GRS validation studies were performed in three previously published case-control cohorts that all included healthy controls (Stanescu et al. NEJM 2011)¹¹.

We then performed additional validation analyses in the NEPTUNE study¹², a prospective cohort study of incident patients with a biopsy diagnosis of MN, FSGS, MCD, or IgAN. This cohort was added to further test if the PLA2R Ab test, GRS, and CRS can distinguish MN from cases of other glomerular diseases that are known to have similar clinical presentations. The results for NEPTUNE analyses are also provided separately, including for a number of subgroup analyses by disease type and ancestry (see Tables S18 for Europeans, S20 and S21 for all other ancestries). We have now moved the description of the validation cohorts from the Supplement to the main methods section and provided more detail on our validation strategy.

To perform comparable analyses of serological test performance in the discovery studies, we also used additional diseased controls with a biopsy diagnosed glomerular disorders across the same disease spectrum as NEPTUNE (FSGS, MCD, and IgAN). Similar to NEPTUNE, these patients were recruited at the time of renal biopsy and were over the age of 18 at the time of study enrollment. The inclusion of extra diseased controls was important in order to demonstrate that the serological and genetic tests can discriminate MN cases from other glomerular disease types with similar clinical features at the time of presentation.

In our sensitivity analyses, we observed no major differences in the performance of serologic and genetic tests when MN cases were tested against healthy versus diseased controls, including by individual disease control type (see Table S18 for details). We now revised the methods and hope that our changes adequately address the reviewer's request.

Title is not accurate and should be modified.

We have now changed the title to “Genetic architecture of membranous nephropathy and its diagnostic implications”, but we are open to additional suggestions by the editors.

Are there differences in MN prevalence across ancestries to justify using just samples of European and East Asian ancestry? If so, please include this in the introduction. Also explain why other ancestries were not included.

As in most genetic studies of rare disease, in this study we were limited to the existing collections of samples from patients with a biopsy diagnosis of MN. Thus, the composition of our discovery cohorts largely reflects the demographics of the centers that collected qualifying DNA samples for genetic studies over the past 15 years. MN appears equally rare across all ancestries, but diagnostic kidney biopsies are not routinely performed in many under-resourced settings (e.g. in Africa or in Central Asia), limiting access to adequately phenotyped cohorts from these regions. We recognize that our cohorts are not fully representative of all ancestries and we emphasize the need for additional studies in other ancestral groups in the discussion. We now also add an explanatory statement to the introduction.

Page 5: under “Study design”, sentence “Eight cohorts were genotyped with high density SNP arrays, imputed using the latest whole genome sequence reference panels, and meta-analyzed genome-wide”. Be specific on the arrays and reference panels. I think this information should be added to the main text and more specific details included in supplementary material.

This was done because of space constraints, but we have now added information on the specific platforms and reference panels used for imputation to the main Table 1, so this information is more accessible to the reader.

Page 6, 2nd paragraph related to complexity of association at the HLA region. Describe differences in the LD structure across ancestries. Some of the results suggest this. Same page, what are “two- and four-digit classical HLA alleles”?

We have added a sentence about known ancestral differences in linkage disequilibrium patterns in this region. To improve clarity, we also changed the term “two- and four-digit classical alleles” to “classical HLA alleles imputed at two- and four-digit resolution”. We also added the following clarification: “The first two digits specify a group of HLA alleles known as super-types, and are defined by older typing methodologies. The third through fourth digits specify nonsynonymous substitutions.”

Page 8, under PLA2R1 locus and its genetic interactions. Please include the N cases/controls by ancestry after the OR. Some of this information is in Table 1 but it is unclear if the study used all the data for analyses (or if there were exclusions based on covariates). I also notice that East Asian ancestry effect estimates had wide confidence intervals suggesting a less precise estimate.

We now include this information as requested. As described in the methods, we were able to perform classical HLA imputation in six discovery cohorts. In total, 2,759 East Asians (803 cases and 1,956 controls) and 4,507 Europeans (1,880 cases and 2,627 controls) with complete classical HLA allele data were included in the detailed analysis of HLA haplotype interactions. We have now added these numbers to the methods and to the legend for Figure 3.

How the gene-gene interaction thresholds were estimated? Provide this information in Methods. Which HLA variants were used given differences among ancestry and more than one HLA independent signal at the HLA region. How many tests were performed for the interaction analyses? Please clarify this.

To screen for interactions, we tested the lead SNPs at each of the 3 non-HLA loci against each of the 5 genome-wide significant independent HLA SNPs (3 in Europeans and 2 in East Asians) resulting in a total of 15 independent tests. We additionally tested for all pairwise interactions between the 3 non-HLA loci shared between both ethnicities (3 additional pairwise tests). Given a total of 18 pairwise interaction tests, we used a conservative Bonferroni-corrected significance threshold of $0.05/18 = 2.8 \times 10^{-3}$. In secondary analyses, we explored which individual HLA risk haplotypes are driving the significant interactions with the *PLA2R1* risk allele without additional corrections for multiple testing. We note that this secondary analysis was performed in a smaller dataset of the six cohorts with fully imputed classical HLA alleles. This information is now included in the methods.

For the sentence " with the risk homozygosity at both loci associated with 89-fold increased odds of disease in East Asians (OR=88.8, 95%CI:38.0-207.3, Pint= 7.8×10^{-3}) and 14-fold in Europeans (OR=14.1, 95%CI:10.0-22.1, Pint= 6.4×10^{-5})". Please add the reference genotypes for these results, i.e., compared to xxx.

We have now added reference genotypes and the exact case/control counts for each genotype class comparison. Please note that wide confidence intervals are explained by the comparison of two relatively rare genotype groups: the double risk homozygotes (as the test group) versus the double protective homozygotes (as the reference group).

Page 9, "Our tissue-specific functional scoring method for non-coding variants prioritized another variant in intron 1, rs17241973 ($r^2=0.93$ with rs17831251) that intersects a putative enhancer element across multiple tissues", please include information specific to kidney tissue, if the disease affects only kidneys.

Adult kidney tissue compartments have not been profiled by the ENCODE or Roadmap Epigenomics consortia, thus kidney-specific data were not available through these resources. This subsequently motivated our own mapping of potential regulatory elements in kidney glomeruli and tubules as presented in Figures S8 and S10. We now provide this clarification in the main text.

Were secondary causes of MN excluded from cases?

Any suspected secondary cases due to drugs, malignancy, infection, or autoimmune disease were excluded. We have added this sentence to the study design section.

Page 10, paragraph on suggestive GWAS loci does not contribute much to the paper and could be removed or moved to the supplementary material.

We have removed this paragraph.

Page 11, which SNPs were included in the GRS? Please include them in the text. Were all the SNPs present in all studies?

The SNPs included in the GRS calculation are now listed in the text; they were either genotyped or imputed in all cohorts.

Page 11, under clinical correlations. Were there any differences by MN disease outcome such as remission/response to therapy?

We agree that this is an important clinical question, but our study was neither designed nor powered to answer it. We have insufficient information on long term outcomes and remission rates for most of our cohorts. This question will need to be addressed by future large prospective studies of primary MN.

Page 12, for the GRS validation in the European-American NEPTUNE GWAS of incident nephrotic syndrome, please clarify the proportion of MN patients in this study and the reasons for testing a GRS derived from MN in a study of nephrotic syndrome that has other etiologies.

We purposefully selected the cohort of incident nephrotic syndrome for additional validation studies, because this cohort enables us to address the most common clinical question in the diagnosis of MN. Primary MN classically

presents with nephrotic syndrome and the common clinical question is how to distinguish incident nephrotic syndrome due to MN from other most common causes of nephrotic syndrome, which includes predominantly FSGS, MCD and IgAN. This motivated the addition of individuals diagnosed with these conditions to serve as diseased controls for the combined serologic and genetic tests in our study. Among 475 NEPTUNE participants, there were 89 cases with the pathology (i.e. biopsy) diagnosis of primary MN and 386 disease controls (184 with FSGS, 164 with MCD, and 38 with IgAN). We feel that the addition of the NEPTUNE analyses provides a unique strength to our study. We have now moved the description of this cohort from the Supplement to the methods section.

Page 13, Neptune GWAS non-European individuals, same as above. What proportion of individuals had MN and how controls were chosen?

There were 133 NEPTUNE participants of African American ancestry (18 cases and 115 disease controls) and 94 participants of Hispanic American ancestry (18 cases and 76 disease controls). For both cohorts, the disease controls included NEPTUNE participants with a biopsy-defined FSGS, MCD or IgAN of the same genetic ancestry as the cases. The numbers of NEPTUNE participants in other ancestral groups were too small for a meaningful analysis. The exact numbers of included individuals are now summarized in the revised Methods and included in the revised Supplemental Tables S20 and S21.

Page 14, 3rd paragraph. The concordant effect of risk loci for MN and IBD is introduced for first time in the discussion. You may want to show some results in the Results section. Also, are these diseases related clinically? This paragraph seems out of place.

These analyses were included in Figure 4a and Table S13, and we have now highlighted these observations in the results section. We have also rephrased the way we discuss these results at the request of the second reviewer (see above).

Page 15, the comparison of genotypes with ELISA test: is the ELISA test standardized across studies? (same test used in all individuals?). How many individuals were used in these analyses?

The same standardized ELISA test is used to test sera across all individuals and studies; this is a commercial ELISA kit manufactured by EUROIMMUN Medizinische Labordiagnostika AG that is currently approved for clinical use in the US, Europe and East Asia. In total, we determined antibody levels in N=2,331 study participants with available sera and genetic data (N=1,488 cases, N=300 healthy controls, and N=543 disease controls). We note that this is the largest study of diagnostic utility of serum anti-PLA2R antibody levels in primary MN. The exact breakdown between discovery and validation cohorts is now described in the methods section on PLA2R antibody testing.

Table 1, were the controls individuals with CKD without MN or individuals without CKD? Add some information to the footnote of the table.

We have added this clarification in the footnote of Table 1.

References:

1. Xu X, Eales JM, Akbarov A, et al. Molecular insights into genome-wide association studies of chronic kidney disease-defining traits. *Nat Commun* 2018;9:4800.
2. Lake BB, Chen S, Hoshi M, et al. A single-nucleus RNA-sequencing pipeline to decipher the molecular anatomy and pathophysiology of human kidneys. *Nat Commun* 2019;10:2832.
3. Pattaro C, Teumer A, Gorski M, et al. Genetic associations at 53 loci highlight cell types and biological pathways relevant for kidney function. *Nat Commun* 2016;7:10023.
4. Wuttke M, Li Y, Li M, et al. A catalog of genetic loci associated with kidney function from analyses of a million individuals. *Nat Genet* 2019;51:957-72.
5. Mladkova N, Kiryluk K. Genetic Complexities of the HLA Region and Idiopathic Membranous Nephropathy. *J Am Soc Nephrol* 2017;28:1331-4.
6. Chen J, Zheng H, Bei JX, et al. Genetic structure of the Han Chinese population revealed by genome-wide SNP variation. *Am J Hum Genet* 2009;85:775-85.
7. Gharavi AG, Kiryluk K, Choi M, et al. Genome-wide association study identifies susceptibility loci for IgA nephropathy. *Nat Genet* 2011;43:321-7.
8. McGrogan A, Franssen CF, de Vries CS. The incidence of primary glomerulonephritis worldwide: a systematic review of the literature. *Nephrol Dial Transplant* 2011;26:414-30.
9. Patterson N, Price AL, Reich D. Population structure and eigenanalysis. *PLoS Genet* 2006;2:e190.

10. Wunnenburger S, Schultheiss UT, Walz G, et al. Associations between genetic risk variants for kidney diseases and kidney disease etiology. *Sci Rep* 2017;7:13944.
11. Stanescu HC, Arcos-Burgos M, Medlar A, et al. Risk HLA-DQA1 and PLA(2)R1 alleles in idiopathic membranous nephropathy. *The New England journal of medicine* 2011;364:616-26.
12. Gadegbeku CA, Gipson DS, Holzman LB, et al. Design of the Nephrotic Syndrome Study Network (NEPTUNE) to evaluate primary glomerular nephropathy by a multidisciplinary approach. *Kidney Int* 2013;83:749-56.

Reviewers' Comments:

Reviewer #1:

Remarks to the Author:

The authors have submitted a very thorough revision and provided high quality responses to all my comments - the scientific value of the paper has been improved by this revision.

Reviewer #2:

Remarks to the Author:

I appreciate the authors' efforts to address the comments raised by me and two other reviewers. The authors have answered most of my questions, while at the same time, some of the information in the response letter was not included in the revised manuscript that could be beneficial for the readers.

My detailed comments are listed below:

- The authors have taken my suggestion to perform the HLA imputation of the Han Chinese cohorts using *Zhou et al.* and reported their findings in the response letter. However, they did not incorporate this change in the main manuscript which I think they should have done given the new imputation panel is better. At minimum, the new results should be included in the supplementary information.
- If the DRB1:1501 association is well known, I suggest the authors to change their wording from "Surprisingly" to "Agreeing with previous findings [refs]" on Line 151 Page 7 to avoid confusion and credit previous work accordingly.
- Please list the high LD regions during QC used in the study in the supplement for reproducibility.
- I strongly encourage the authors to include the global PCs in response to my comment 4(b) as a SF to give readers a better view of the population structure included in the study (preferably with 1000 Genomes samples colored correspond to their continental population).
- The authors have now performed a fine-mapping analysis, in addition to epigenetic data supporting the potential causal variants in their description of the PLA2R1, NFKB1 and IRF4 loci, please can the authors also state the posterior probability for each variant to better inform the readers.
- IBD was never defined in the manuscript. Are the pleiotropic effects mostly from UC or both UC and CD?
- Linkage disequilibrium (LD) was introduced first on Page 6 Line 124 instead of on line 158, Page 7.
- Lack of reference for HLA reference panels on Line 135, Page 7.

Reviewer #3:

Remarks to the Author:

The authors have answered all my questions. A minor issue is the need to include the reference population used for the LD in trans-ethnic analysis plots shown in Table 1.

We provide additional point-by-point responses to the reviewer comments below (in bold font):

Reviewer #1 (Remarks to the Author):

The authors have submitted a very thorough revision and provided high quality responses to all my comments - the scientific value of the paper has been improved by this revision.

We appreciate these kind comments by the reviewer.

Reviewer #2 (Remarks to the Author):

I appreciate the authors' efforts to address the comments raised by me and two other reviewers. The authors have answered most of my questions, while at the same time, some of the information in the response letter was not included in the revised manuscript that could be beneficial for the readers. My detailed comments are listed below:

- The authors have taken my suggestion to perform the HLA imputation of the Han Chinese cohorts using *Zhou et al.* and reported their findings in the response letter. However, they did not incorporate this change in the main manuscript which I think they should have done given the new imputation panel is better. At minimum, the new results should be included in the supplementary information.

These were performed as a technical validation of our main findings, thus we provided them in the Methods section including all relevant analytical methods and references.

- If the DRB1:1501 association is well known, I suggest the authors to change their wording from "Surprisingly" to "Agreeing with previous findings [refs]" on Line 151 Page 7 to avoid confusion and credit previous work accordingly.

The association of DRB1*1501 has been reported previously in Chinese, but the effects of this allele in Europeans have not been examined previously, so this finding is new and surprising. We would like to keep the current text unchanged.

-Please list the high LD regions during QC used in the study in the supplement for reproducibility.

This information has now been added to the Supplementary methods with a corresponding reference.

-I strongly encourage the authors to include the global PCs in response to my comment 4(b) as a SF to give readers a better view of the population structure included in the study (preferably with 1000 Genomes samples colored correspond to their continental population).

This information has now been added as a Supplementary Figure 2.

- The authors have now performed a fine-mapping analysis, in addition to epigenetic data supporting the potential causal variants in their description of the PLA2R1, NFKB1 and IRF4 loci, please can the authors also state the posterior probability for each variant to better inform the readers.

We have added this information to the legend of the Supplementary Figure 5, and we also included a posterior probability of signal co-localization between East Asian and European meta-analyses.

- IBD was never defined in the manuscript. Are the pleiotropic effects mostly from UC or both UC and CD?

Thanks for pointing this out – to avoid ambiguity, we have now defined IBD as both ulcerative colitis and Crohn's disease in the legend for Figure 5. We have also added definitions for PBC, GFR, and HBV that were previously missing.

- Linkage disequilibrium (LD) was introduced first on Page 6 Line 124 instead of on line 158, Page 7.

Thanks, this is now corrected.

- Lack of reference for HLA reference panels on Line 135, Page 7.

When mentioning reference panels, we now added "(see Methods)", where these reference panels are described.

Reviewer #3 (Remarks to the Author):

The authors have answered all my questions. A minor issue is the need to include the reference population used for the LD in trans-ethnic analysis plots shown in Table 1.

We assume the reviewer refers to the conditional plots in Figure 1. We did not use LD reference in the trans-ethnic conditional analyses, since we conditioned on the top SNP in each individual cohort, then meta-analyzed conditioned summary statistics as described in the Methods. We have now added a clarifying statement to the figure legend.